# Finding differentially expressed sRNA-Seq regions with srnadiff

**Matthias Zytnicki***, **Ignacio González**

INRAE, MIAT, Toulouse, Castanet, France

* matthias.zytnicki@inra.fr

## Abstract

Small RNAs (sRNAs) encompass a great variety of molecules of different kinds, such as microRNAs, small interfering RNAs, Piwi-associated RNA, among others. These sRNAs have a wide range of activities, which include gene regulation, protection against virus, transposable element silencing, and have been identified as a key actor in determining the development of the cell. Small RNA sequencing is thus routinely used to assess the expression of the diversity of sRNAs, usually in the context of differentially expression, where two conditions are compared. Tools that detect differentially expressed microRNAs are numerous, because microRNAs are well documented, and the associated genes are well defined. However, tools are lacking to detect other types of sRNAs, which are less studied, and whose precursor RNA is not well characterized. We present here a new method, called srnadiff, which finds all kinds of differentially expressed sRNAs. To the extent of our knowledge, srnadiff is the first tool that detects differentially expressed sRNAs without the use of external information, such as genomic annotation or additional sequences of sRNAs.

**Data Availability Statement:** The tool presented in the paper is freely available through BioConductor. Published datasets are available from these GEO ids: H. sapiens: GSM927308 GSM927309 GSM927310 GSM927311 GSM927312 GSM927313 GSM927314 GSM927315

## Introduction

The interest in small RNA (sRNA) has been constantly growing since the last decades. These sRNAs include the well-known microRNAs, but also tRNA-derived RNA fragments (tRFs), small interfering RNAs (siRNAs), Piwi-associated RNAs (piRNAs), small nucleolar RNAs (snoRNAs), which have been extensively studied in plants and animals (see [1, 2] for reviews on sRNAs on both kingdoms). The majority of these RNAs are less than 50 bp, but some may reach up to 200 bp. These sRNAs are involved in many stages of development and diseases, in genetic and epigenetic pathways. The regulation of the expression induced by these sRNAs is usually understood *via* a differential expression protocol, *e.g.* healthy *vs* sick, or wild type *vs* mutant.

Sequencing these sRNAs is now the state-of-art technique in order to study them. It accurately quantifies all the sRNAs in a single experiment –known and unknown sRNAs alike– even though it suffers from some known biases [3]. However, most analyses only focus on miRNAs, partly because miRNAs genes (pre-miRNAs or mature miRNAs) are known, or can be efficiently discovered.

GSM927316 GSM927317 GSM927318
GSM927319 D. melanogaster: GSM2042871
GSM2042873 GSM2042875 GSM2042877
GSM2042879 GSM2042881 GSM2042883
GSM2042885 GSM2042887 GSM2042889
GSM2042891 GSM2042893 D. melanogaster:
GSM2042871 GSM2042873 GSM2042875
GSM2042877 GSM2042879 GSM2042881
GSM2042883 GSM2042885 GSM2042887
GSM2042889 GSM2042891 GSM2042893 A.
thaliana: GSM906549 GSM906550 GSM906551
GSM906552 The synthetic dataset has been made
available at the Data INRAE repository. The
permanent link is: https://doi.org/10.15454/
PA1CW6 The preprocessed data has been made
available at the Data INRAE repository. The
permanent link is: https://doi.org/10.15454/
0DCIGO.

**Funding:** The author(s) received no specific
funding for this work.

**Competing interests:** The authors have declared
that no competing interests exist.

In this work, we will suppose that a reference genome is available. Although some organisms are still not assembled, most model and many non-model organisms are available, and the sequencing of the remaining organisms progress at a fast pace.

Contrary to messenger RNA-Seq analysis, there is, to the best of our knowledge, no recent review, nor guideline, on sRNA-Seq differential expression analysis. However, we observed that current papers usually use three types of pipe-lines. First, the reads can be mapped to a set of reference sequences, such as miRBase [4], which stores all the known miRNAs. The counts are then arranged into a matrix, where the rows are the features (here, the miRNAs), the columns are the samples, and the cells are the number of reads that match a given feature, in a given sample. This matrix is then used as an input of the standard (messenger) RNA-Seq protocol for differential expression, where differential expression is tested. Many tools have been presented to perform this last step, but DESeq2 [5] and edgeR [6] are, by far, the most cited ones.

This method has been used, for instance, by [7], who analyzed the sRNA-Seq data of lung tumors compared to adjacent normal tissues. This method can obviously only find features that are previous detected, and usually restrict the analysis to only one, or few, classes of small RNAs.

The second option has been for instance used by an other article, that reused the previous dataset in order to find differentially expressed snoRNAs and piRNAs [8]. Here, the authors followed the usual pipe-line of the messenger RNA-Seq. They mapped the reads to the genome, and compared the mapped reads with external annotations, here piRNAs (from piRNABank [9]), and snoRNAs (from UCSC genome browser annotation [10]). The authors claimed that the approach is more exhaustive, since —especially in human— the annotation files which are provided by the existing repositories include a wide diversity of small RNAs.

While this approach works fine on some well characterized sRNAs, such as miRNAs, tRFs, or snoRNAs, they are much less efficient for others, such as siRNAs or piRNAs, because the corresponding "genes" are not known (and possibly even not defined). Indeed, current piRNA databases are usually restricted to human and mouse, and their accuracy is not well established.

Some popular tools for sRNA differential expression, such as UEA sRNA Workbench [11] and sRNAtoolBox [12] use a combination of the two previous methods. While these method works fine for miRNAs, and other well-know sRNAs, they cannot detect other types of sRNAs, or be applied to an unannotated genome.

It is, at least in principle, always possible to complement the annotation of the genome of interest using tools that detect a class of small RNAs *de novo*: tRNAscan-SE [13] for tRNAs, Snoscan [14] for snoRNAs, or infeRNAl [15] for small RNAs described in the RFAM database [16] are the most widely used tools. However, this requires expertise, as well as computational resources. It is, in practice, rarely done, and still restricts the analysis to well-characterized small RNAs.

It is also possible to cluster unknown, co-localizing reads to putative sRNAs. However, finding boundaries of sRNAs based on the expression profile is a difficult task, because the expression profile of the sRNAs can be very diverse: miRNAs and tRNAs, for instance, exhibit sharp peaks, whose sizes are approximately the size of a read. The expression profiles of siRNAs and piRNAs are usually wider, ranging possibly several kilo-bases or more, with an extremely irregular contour. These sRNAs can even be found in clusters, and the aggregation of several, possibly very different, profiles, makes it hard to discriminate them.

Some methods, such as BlockClust [17], SeqCluster [18], or ShortStack [19], do include a clustering step, which assemble the reads into longer transcripts. Then, the user can proceed to the standard messenger RNA-Seq pipe-line: counting reads that co-localize with each

transcript, and testing for differential expression. These tools may or may not use an annotation file. The downside of this approach is that it requires significantly more work and time to cluster the reads into putative transcripts. Moreover, the produced regions may have no biological meaning since they may be truncated, or merged transcripts.

Recently, [20] presented derfinder, a new method for discovering differentially expressed (long) genes. Briefly, the authors find differentially expressed regions at the nucleotide resolution, regardless of the annotation. This promising method, however, has been designed for RNA-Seq and works poorly on sRNA-Seq, because sRNA expression profiles are very different from the longer, somewhat uniform, expression profiles of the exons.

In this work, we present a new method, srnadiff, that finds differentially expressed small RNAs, using RNA-Seq data alone, without annotation. We show that srnadiff is more efficient than other methods, and that it detects a wide range of differentially expressed small RNAs.

## Materials and methods

### Description of the method

The method is divided into two main steps, which are described hereafter. The outline of the method is given in Fig 1. Briefly, the first step includes several independent methods that detect putative differentially expressed regions. The second step considers all the regions found by the previous step, and chooses the best ones.

**Step 1: Find candidate regions.** In this step, several methods are used to produce genomic intervals that are potential differentially expressed regions. We implemented three methods:

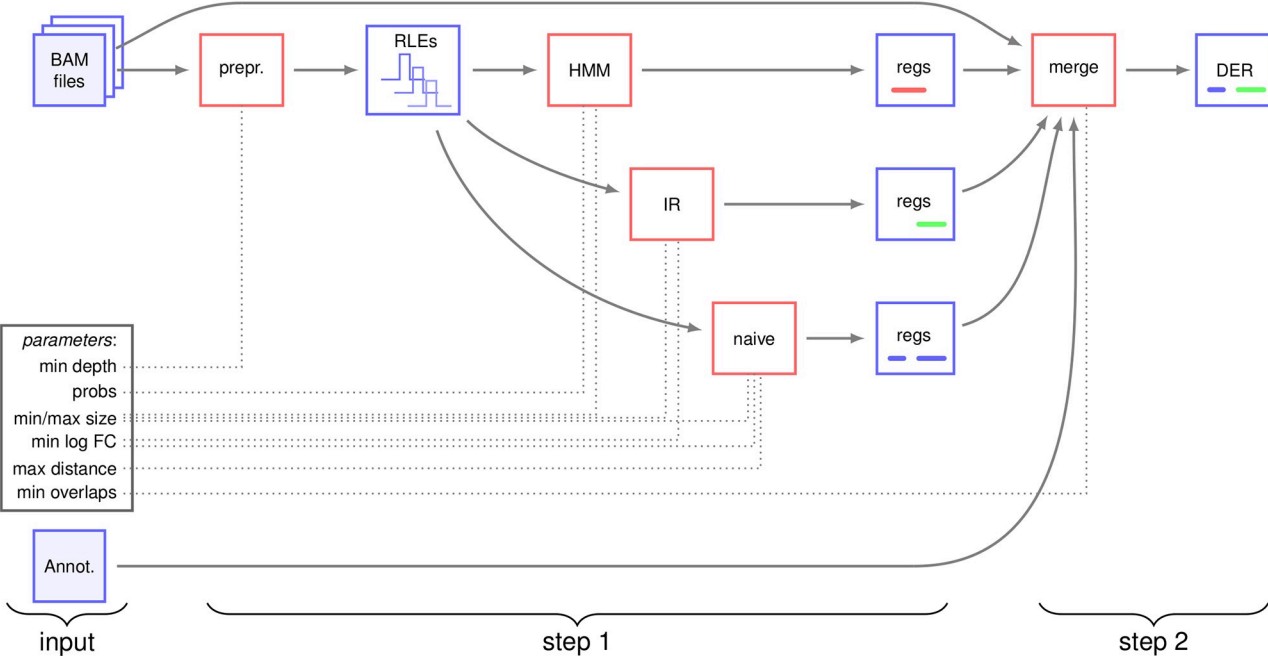

**Fig 1. Outline of the method.** Files and intermediate data are displayed in blue rectangles, algorithms are displayed in red rectangles, and parameters are in the black rectangle. Arrows show the data which are used by each algorithm. The first step transforms BAM files into coverages, stored as run-length encoding. The coverages are then used by different methods (naive, HMM, IR), which independently produce differentially expressed regions, stored are lists of genomic intervals. The user can also add an annotation, which is transformed into putative regions. In a second step, these regions are merged. These regions may overlap, since they have been predicted by several, independent tools. We compute the p-values of the differential expression, and, in case of overlap, choose the regions with the lowest p-values. The parameters (described hereafter) used by each algorithm are in shown dotted lines. prepr.: pre-processing; RLE: run-length encoding: regs: regions; DER: differentially expressed regions.

naïve, HMM, and IR (described in the next sections), but, in principle, any method can be added.

Indeed, expression profiles are very specific to each of the sRNA classes: miRNAs pile up in short stacks, whereas piRNAs spread over thousand of nucleotides. In order to cope with this diversity, we found that two different methods yielded better results than a unique one. The HMM method efficiently detects short, moderately differentially expressed RNAs. The IR method can detect longer patterns.

Each method reads a set of coverages (see next section), and outputs a list of genomic intervals, encoded as `GenomicRanges`[21].

**Preprocessing.** Prior to the analysis of the data, the samples are first normalized using the CPM procedure, as done in edgeR [6].

Moreover, the following strategies use a run-length encoding representation of the data, which is a compact way to store the expression of each nucleotide of the genome. This process is described in Fig 2A.

If the average depth is less than a given threshold (`min depth`, default: 10) in both conditions, the corresponding position is discarded and will not be considered in the future analyses.

**Annotation.** This step simply provides the intervals corresponding to the annotation file that is optionally given by the user. It can be a set of known miRNAs, siRNAs, piRNAs, or a combination thereof.

**Naïve.** The outline of the method is shown in Fig 2B. This strategy computes the average expression for each condition. Then, the (log2) fold change of the expression is computed. All the regions with a fold change greater than the parameter `min log fold change` are kept as putative regions. The putative regions that are distant by no more than `max distance` nucleotides are then merged. However, we do not merge two regions if their log2 fold change have different signs. The remaining intervals are provided as candidate regions.

The parameters used by this methods are: a minimum log fold change (`log fold change`, default: 0.5), a maximum merged distance (`max distance`, default: 100) and minimum and maximum region sizes. Regions whose sizes do not fit the aforementioned bounds (`min size`, `max size`, default: 18 and 100,000) are discarded.

**HMM**. We first build a matrix, where each line is a nucleotide, each column a sample, and each cell is the corresponding expression. This matrix is given to a differential expression analysis package, such as DESeq2. We then proceed to the standard differential analysis workflow, and we compute an adjusted p-value for each nucleotide (see Fig 2C).

This method is expected to give the same p-value for each nucleotide of a differentially expressed miRNA or tRF than the annotate-then-identify method. Contrary to messenger RNAs, the reads spans the whole transcript. Supposing that a sRNA, such as a mature miRNA, is sequenced $x$ times, the annotate-then-identify will quantify an expression of $x$ for this feature. Our method will also find the same expression of $x$, because each nucleotide has a sequencing depth of $x$. The only difference is that we will likely have identical counts for each nucleotide of the miRNA. This is why, in our implementation, we collapse lines of the count matrix that have exactly the same counts into a unique row, because the counts actually describe the same object.

The use of DESeq2 in this context is obviously not standard, because DESeq2 expects the total number of reads per transcript, and not the nucleotide-wise number of reads. However, in an ideal case of a miRNA, tRF, or snoRNA, the counts found by this method and the usual one are identical, as noted previously. We also plotted several quality control distribution, and checked that they behave as expected. First, we plotted the dispersion estimation. Briefly, the read counts for a particular gene are expected to follow a negative binomial distribution, with

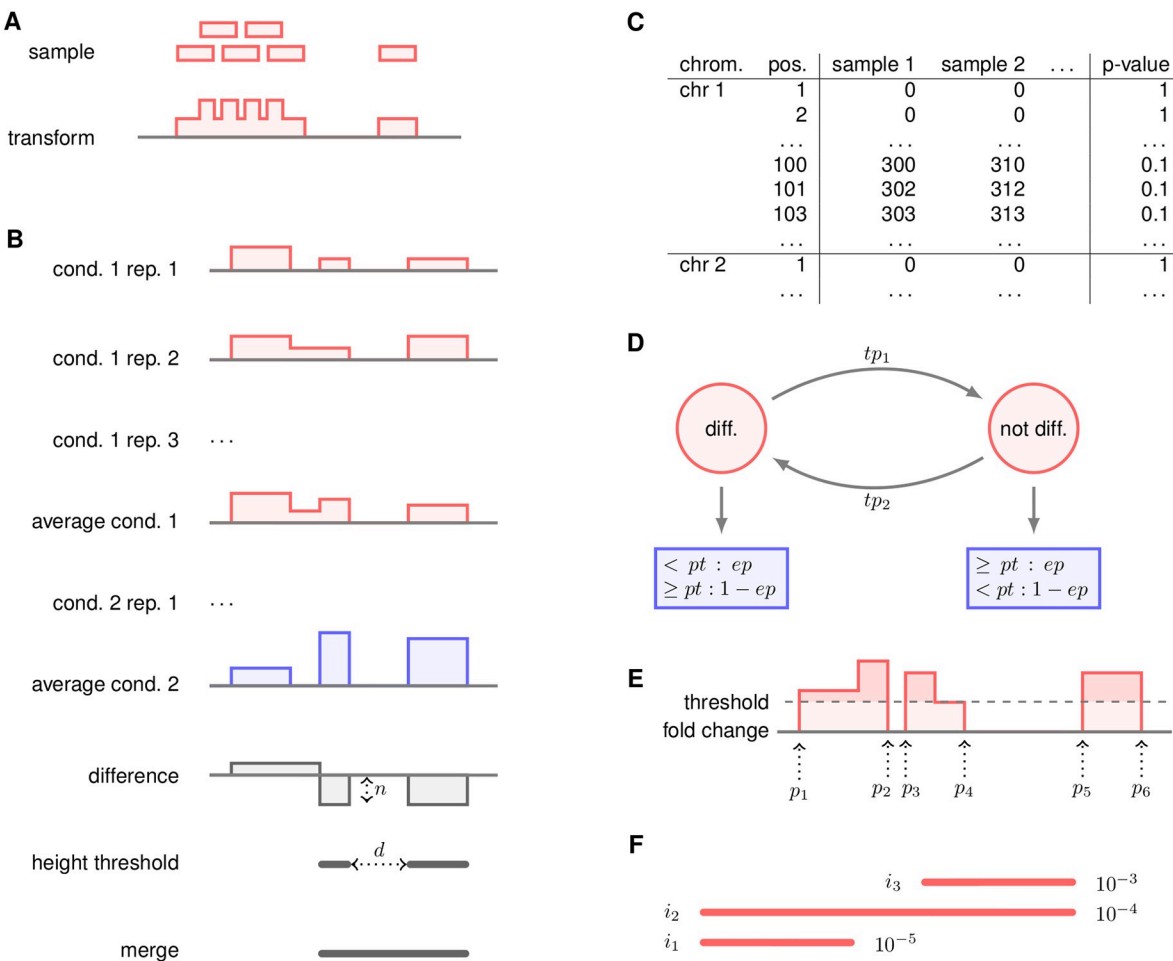

**Fig 2. Description of the steps. A**: Transformation from mapped reads to run-length encoding. The reads themselves are lost, only the coverage is kept. For the sake of memory compactness, the coverage is stored as a vector of pairs (coverage, length) per chromosome. **B**: The naïve method. In a first step, the samples are averaged for each condition, then the (log2) fold change is computed. All those regions with a fold change not less than $n$ are kept. Regions distant than no more than $d$ base pairs are then merged, and given as output of the method. **C**: First step of the HMM method. The coverage of each nucleotide is computed for every condition. A p-value is produced for each position. **D**: Second step of the HMM method. An HMM is run on each chromosome. The states are the red circles, and the emission probabilities are the blue rectangles. diff: the "differentially expressed" state. not diff: the "not differentially expressed" state. $tp_1$ and $tp_2$ are the transition probabilities. The emission of each state follows a binomial distribution. For instance, the diff. state emits a p-value less than $pt$ with probability $ep$. All the parameters ($tp_i$, $pt$ and $ep$) are editable by the user. **E**: IR step. The (log2) fold change is printed in red, 0 is given in solid grey, and the dashed line is the (user given) threshold. Every region above the threshold is a putative differentially expressed region. A simple method could give three regions (between $p_1$ and $p_2$, between $p_3$ and $p_4$, and between $p_5$ and $p_6$). The IR method aims at merging close-by regions, with no additional parameter (contrary to the naïve method). Briefly, the method considers every interval ($p_1$–$p_6$, for instance), computes the area above the threshold (in light red), and divides it by the size (here, $p_6 - p_1$). We will call the mean area above the threshold MAAT. The method then considers all the positions, for instance $p_3$, between $p_1$ and $p_6$. If the MAAT between $p_1$ and $p_3$, or the MAAT between $p_3$ and $p_6$, is not above the threshold, the interval is split. In the example, the MAAT between $p_3$ and $p_6$ is visibly less than the threshold, so the region is split at $p_3$. However, the region between $p_1$ and $p_4$ is not split, since the MAATs between $p_1$ and $p_2$, $p_1$ and $p_3$, $p_2$ and $p_3$, $p_2$ and $p_3$ are all greater than the threshold. **F**: The merge step. Each interval $i_1$ to $i_3$, is associated to a p-value, written on the right of the interval. A naïve approach would discard $i_2$ and $i_3$ because they are dominated by $i_1$ and $i_2$ respectively. However, $i_3$ may be an interesting interval, although the signal is not as strong as the signal of $i_1$. We can notice that $i_2$ both dominates ($i_3$) and is dominated (by $i_1$). Only this interval is discarded.

two parameters: the mean, and the dispersion. The dispersion is estimated using the mean, and is expected to follow a trend for the majority of the genes. If many genes depart from the trend, the expression is clearly poorly estimated. In Fig 1a in S1 Appendix, we show that, on the human dataset, the dispersion plot is close to the expected one. This confirms that the

estimation step was correctly carried out. Second, we plotted the p-value distribution. The p-value is expected to be uniform under the null hypothesis, and close to 0 under the alternative hypothesis. The Fig 1b in S1 Appendix confirms that we observe a mixture of the two distributions on the human dataset.

In a second step, we build a hidden Markov model (HMM) on each chromosome, where the first state is "differentially expressed," the second state is "not differentially expressed," and the observations are the p-values (see Fig 2D). This HMM has been given sensible emission, transition, and starting probabilities values, but these parameters can be tuned by the user (S1 Appendix shows that the method does not seem sensitive to parameters). We then run the Viterbi algorithm, in order to have the most likely sequence of states. The regions that are most likely to be in the "differentially expressed" state are given as output of the method.

In practice, the p-value is not computed for every nucleotide. Regions where the sum of the coverage is less than a threshold (10 by default, editable by the user) are given a p-value of 1, because these poorly expressed regions are unlikely to provide a differentially expressed sRNA. In the HMM, all the regions with a p-value of 1 (the majority of the genome, because small RNA transcription is restricted to a minority of loci) are not stored and are assumed to have the default value. This significantly reduces the memory consumption. During the Viterbi algorithm, the most frequent state is "not differentially expressed," and the most frequent p-value is 1. In this configuration, if the probability of "not differentially expressed" is significantly larger than the probability of the other state, we directly skip to the next nucleotide with a p-value $< 1$. Indeed, the difference of the probabilities of the "not differentially expressed" and "differentially expressed" states are, in this case, constant, and do not change the results the Viterbi algorithm.

We implemented this Viterbi algorithm, optimized for our purposes, in C++.

The default values used by this method (see Fig 2D) are $tp_1 = 0.001$, $tp_2 = 0.000001$, $pt = 0.1$, $ep = 0.9$. This method also uses size bounds, akin to the previous step.

**Irreducible regions.** The average (normalized) coverage of each condition is first computed. We then compute the (log2) fold change, and find irreducible regions (IRs), as presented in [22]. The method is presented in Fig 2E. Briefly, the method extracts all the regions where the fold change is above a threshold (given by the user). The IR method is a simple and efficient way to merge such regions when they are not very far away, and the drop in fold changed is not too deep.

In practice, the IR method can be very efficiently implemented. It simply requires a linear time algorithm, that considers all the points where the fold change intersects the threshold.

We also take care not to merge regions with positive log fold change, and regions with negative log fold change.

We implemented the algorithm in C++, using the description from [22]. The only parameter is `min log fold change` (default: 0.5).

**Step 2: Merge regions.**   The intervals provided by the previous step may overlap since several methods may give similar intervals. The aim here is to keep only the best non-overlapping regions.

To do so, the intervals provided in the previous steps are used as standard genes and we use the RNA-Seq standard pipe-line.

- The expression of the intervals is quantified for each condition (a read is counted once for every interval it overlaps).

- By default, DESeq2 is used to compute a p-value for each intervals.

When two regions, $i_1$ and $i_2$, overlap, $i_1$ *dominates* $i_2$ iff its p-value is less than the p-value of $i_2$. A first possibility is to give undominated intervals to the user, but we found that it removes many interesting intervals (see Fig 2F).

Our method only discards all the intervals that are both dominated, and dominate other intervals. When these intervals have been discarded, only undominated intervals remain, and they are given to the user (together with their p-values).

This step uses one parameter: the minimum number of common nucleotides between two regions $i_1$ and $i_2$ to consider that $i_1$ and $i_2$ overlap. It outputs a list of differentially expressed regions, as `GenomicRanges`, together with their p-values.

## Implementation of srnadiff

**Example of use.** srnadiff can be installed through R Bioconductor [21]. As every Bioconductor package, it has a dedicated Web page (https://bioconductor.org/packages/release/bioc/html/srnadiff.html) and contains an extensive description of the tool.

We provide here an introduction on how to use the package.

The information about the dataset can be conveniently stored in a `data.frame`. This table should contain three columns, and each row describes a sample. The columns provide the name of the BAM files, the name of the sample, and the condition (e.g. `wild_type` *vs* `mutant`).

If the table is stored into a file named `data.csv`, the minimal code to run srnadiff is:

```
1 library(srnadiff)
2 data        <- read.csv("data.csv")
3 bamFiles    <- file.path(dir, data $FileName)
4 exp        <- srnadiffExp(bamFiles = bamFiles,
5                 sampleInfo = data,
6                 annotReg = "annotation.gtf")
7 exp        <- srnadiff(exp)
8 diffRegions  <- regions(exp, 0.05)
9 plotRegions(exp, diffRegions [1])
```

In this code, `annotationFile.gtf` is a GTF file that contains the known annotation. It is an optional parameter.

The `srnadiffExp` function reads the input data, and transforms the BAM files into run-length encoding data. It returns an object of class `srnadiffExp`.

The `srnadiff` function performs the main tasks of the package: segmentation, reconciliation, and computation of the p-values. The parameters that control the algorithms can be changed using this function. The `segMethod` parameter takes the list of the segmentation methods that should be used (default is "HMM" and "IR"). The `nThreads` parameter controls the number of threads used. The other fine-tuning parameters (such as the minimum sequencing depth, the minimum and maximum feature ranges, etc.) are stored into the `srnadiffDefaultParameters` object. This object can be changed as desired, and provided to the `srnadiff` method.

The `regions` function provides the differentially expressed regions, in a `GenomicRanges` object [23], which can be exported to BED, GTF, or GFF files with the `rtracklayer` package [24]. A minimum (adjusted) p-value can be provided as parameter.

The `plotRegions` function is a utility tool, which plots the coverage of the different samples around a region of interest (usually a prediction of srnadiff). This function accept a great number of parameters to customize the plot (visual aspect, other annotation, etc).

**Computation of the p-values.**   p-values at computed twice in the process. First, they are computed for each position of the genome in the HMM strategy. Second, candidate regions are tested for differential expression, and given a p-value. By default, DESeq2 [5] is used to perform the two steps. However, for the sake of completeness, we made two other widely used packages available for the user: edgeR [6], and baySeq [25]. The alternative package can be chosen by specifying the parameter `diffMethod` in the `srnadiff` function.

## Benchmarking

We benchmarked srnadiff on three real, already published, datasets, and a synthetic one. We selected several datasets meeting the following criteria. First, the experimental design should include replicates in each condition. Second, the differentially expressed elements found by the authors should be accessible as supplementary data of the article. Third, we wanted to include a variety of model organisms: *Homo Sapiens*, *Arabidopsis thaliana* and *Drosophila melanogaster*. Fourth, the sequencing machines should be different, in order to include the diversity of the machines in our benchmark. Last, the analysis pipe-lines should also be different. All the publications provided a list of differentially expressed sRNAs, and we compared the different methods with this list of sRNAs.

Of note, it is not possible to assess, on real-life datasets, the number of false positives, *i.e.* the regions detected by the evaluated tools which are not differentially expressed, because it is not possible to find all the differentially expressed small RNAs. Therefore, metrics such as specificity or precision cannot be collected on these datasets. The aim of the benchmark on published datasets is to test whether the evaluated tools are able to detect all the regions that have been classified as differentially expressed by the reference methods. In order to assess the number of false positives, we added a last, simulated, dataset.

srnadiff was run with no annotation, and an adjusted p-value threshold of 5%. We also run derfinder [20] on the same datasets, with a q-value of 10%. We used a third method, which first clusters the reads with ShortStack [19] (comparing several clustering methods is out of the scope of this article), quantified the expression of the regions found by ShortStack with feature-Counts, tested for differential expression with DESeq2, and kept the regions with an adjusted p-value of at most 5%. We refer to this method as the *ShortStack* method. The reason why we chose a q-value of 10% for derfinder, instead of 5%, is that the statistics produced by derfinder is significantly more conservative, and it produces much fewer predicted regions than other approaches. For a fair comparison, we decided to lower the stringency for this tool. No fold change threshold was applied.

We compared the results of the methods with respect to each other.

We also compared srnadiff with another straightforward method: we downloaded available annotations of different sRNA-producing loci, and followed the previously presented method: expression quantification, and test for differential expression. These regions can be considered as true positives. For the human dataset, miRNAs were taken from miRBase [4], tRNAs (in order to find tRFs) from GtRNAdb [26], piRNAs from piRBase [27], snoRNAs from Ensembl [28], and genes (to find possible degradation products), from Ensembl too. We added new additional databases: tRNAs and snoRNAs from UCSC [10], and piRNAs from https://www.pirnadb.org for an additional validation. We also used miRDeep2 [29] in order to find putative new miRNAs using the sequencing data. The cress data was extracted from the TAIR annotation file [30], and from the FlyBase annotation file [31] for the fly. We then compared these differentially expressed regions with the predicted regions.

We stated that the two predicted regions *A* and *B* were similar when at least 80% of *A* overlaps with *B*, or at least 80% of *B* overlaps with *A*. The reason is that some annotation (such as

genes, or tRNAs which are substentially largers that tRFs) are not expected to be differentially expressed. Moreover, ShortStack provides also significantly larger differentially expressed regions than srnadiff or derfinder.

For each tool, we plotted different results. First, we provided the size and the adjusted p-value distributions of the regions found. Then, we provided the number of differentially expressed features (e.g. differentially expressed miRNAs, tRFs, etc.) which overlap a given region found. The aim here was to test whether a method would "merge" several potential candidates into a unique, longer, differentially expressed region. Then, we focused on the regions found by srnadiff and another tool (derfinder or ShortStack). For each such region, we compared its size, and its (adjusted) p-value found by each method.

The code used for the benchmarking and the versions of the tools used are given in S1 Appendix.

## Datasets

### Preprocessing

Published datasets were downloaded from SRA [32] using the SRA Toolkit. We cleaned the data with fastx_clipper (http://hannonlab.cshl.edu/fastx_toolkit/index.html), mapped them with bowtie [33] (because it was ranked favorably in a recent benchmark [34]).

**Human dataset.** The first dataset compared healthy cells *vs* tumor cells of lungs of smokers. This dataset has been sequenced on a Illumina GA-IIx, and contains 6 replicates per condition, with about 26 millions reads per sample. It has been published by [7], and re-analyzed by [8]. Both papers analyzed the sRNA-Seq data of lung tumors compared to adjacent normal tissues.

In the first paper, the authors mapped the reads with bowtie [33] on miRBase [4]. The miRNAs were tested for differential expressed with edgeR [6].

In the second paper, the authors mapped the reads with Novoalign on the reference genome. They downloaded piRNA and snoRNA annotations, and quantified their expression with the BedTools [35]. They were tested for differential expressed with edgeR.

### A. thaliana dataset

This dataset evaluated the difference of expression of small RNAs in two different concentrations of $CO_2$ in *A. thaliana* [36]. Each condition contained two replicates, with about 11 millions reads each.

The authors mapped the reads with RazerS [37] on the reference genome. Reads co-localizing with known miRNAs were kept for further analysis. The authors then looked for other, unknown putative miRNAs. Reads overlapping with known cDNAs, miRNAs, tRNAs, rRNAs, snoRNAs, and other RNA sequences, were excluded. Remaining reads were then given to miRDeep [38] and miRCat [39], two miRNA finding tools. The new set was added to the previous, known set of miRNAs. Differential expression analysis was performed with a $\chi^2$ test.

### D. melanogaster dataset

In this dataset, [40] sequenced small RNAs of young and aged *D. melanogaster* flies circulating in the hemolymph. Each condition contained 8 and 4 replicates, with an average of 13 millions reads each.

The authors mapped the reads with bowtie on the reference genome. Detection of new miRNAs, and quantification of known and new miRNAs, was performed with miRDeep2, using data from miRBase. Micro RNAs were tested for differential expression with edgeR.

## Synthetic dataset

We also generated a synthetic dataset, extracted from the human genome. We first selected 1000 miRNAs from miRBase [4], and 1000 piRNAs from piRNABank [9]. We randomly selected 100 upregulated miRNAs (2-fold change), 100 downregulated miRNAs (2-fold change), and the same for the piRNAs. We randomly assigned a baseline expression following a power law ($k = 1.5$) on the miRNAs and the piRNAs, which reflected a low expression for most of the RNAs, and a few very expressed RNAs, as observed on our data. We generated 6 replicates per condition using the polyester package [41]. Obviously some RNAs may be differentially expressed, but with a very low expression, they are almost impossible to detect. As a results, we quantified the expression of the features with featureCounts [42], tested for differential expression with DESeq2 [5], and restricted to all the features with an adjusted p-value of at most 5%. We considered these regions as our "truth" dataset.

## Results

### Human dataset

**Comparison to the truth set.**   The first article found 48 differentially expressed small RNAs, and the second, only 23 (see Fig 3). Only 5 miRNAs were common in both analyses.

We first wondered whether we could also detect these differentially expressed miRNAs using our method. Fig 4 shows the recall of the different methods. The recall (or sensitivity) is defined as $\frac{TP}{TP+FN}$, where $TP$ is the number of true positives, and $FN$ is the number of false negatives. In our benchmark, a true positive is region which has been detected as differentially expressed by a reference set (here, the miRNAs published in the articles), such that at least 80% of the nucleotides overlap with a region detected as differentially expressed by srnadiff (or other evaluated tools such as derfinder, and ShortStack). Conversely, the false negatives are the regions which has been detected as differentially expressed by a reference set, which do not overlap (with at least 80% of the nucleotides) with a region detected as differentially expressed by the evaluated tools.

First, the expression of six miRNAs found by [7] could not be correctly estimated because they belonged to duplicated regions in the new assembly (and not in the assembly used in the paper). Second, a miRNA found by [8] was missed because it was considered as an outlier by DESeq2.

We then compared these results with srnadiff (run with no annotation, and an adjusted p-value threshold of 5%). srnadiff finds 1968 differentially expressed regions in total. It missed a few miRNAs, because of an adjusted p-value threshold effect: when the test for differential expression is performed on a few miRNAs (here, 48 and 23 respectively), the adjustment is not expected to change the p-values. However, srnadiff has much more candidates (a few thousands), that should be tested. As a consequence, the adjustment is much stronger in this case, and many miRNAs have an adjusted p-value which is (slightly) greater than 5%. It is a usual trade-off between precision and recall.

Then, we compared our results with derfinder, which finds 76 differentially expressed regions. Most of the regions found by derfinder are also found by srnadiff (see Table 1). srnadiff missed some regions, because of the adjusted p-value threshold effect. Of note, no region

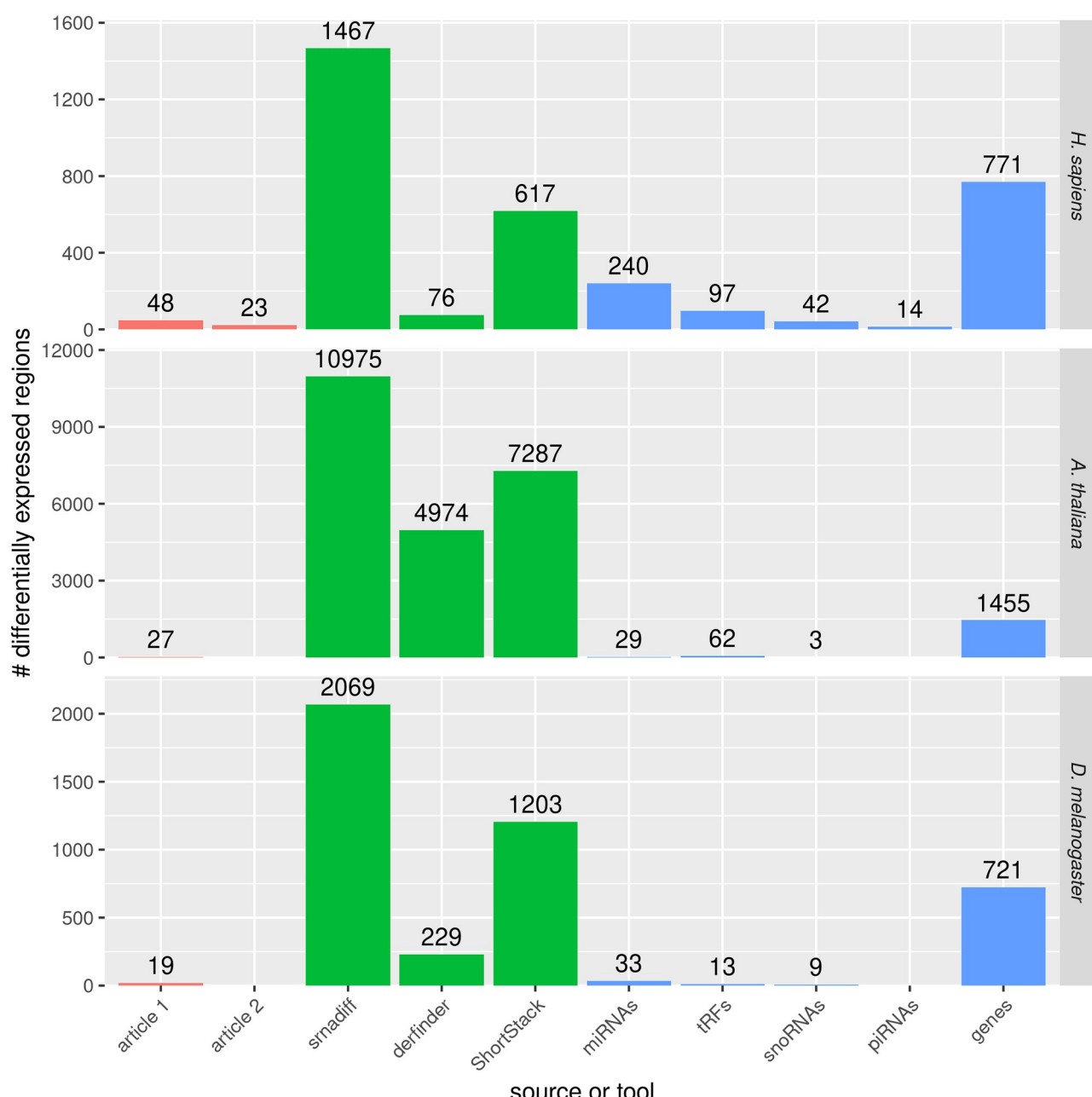

**Fig 3. Number of differentially expressed regions found by each method.** The red bars indicates the number of miRNAs found by [7] and [8] for the *H. sapiens* dataset, [36] for the *A. thaliana* dataset, and [40] for the *D. melanogaster* dataset. The green bars indicate the number of differentially expressed regions found by the three methods which are evaluated: srnadiff, derfinder, and ShortStack. The last bars give the number of elements that are found using the known annotation and the usual differential expression calling method.

found by derfinder with adjusted p-value less than $10^{-5}$ was missed. However, srnadiff provided significantly more regions (see Fig 3).

We then compared with the ShortStack method, which finds 617 differentially expressed regions. Results show that srnadiff misses several regions. The main reason is that ShortStack accepts very large regions that may have a low p-value on the whole region even if the

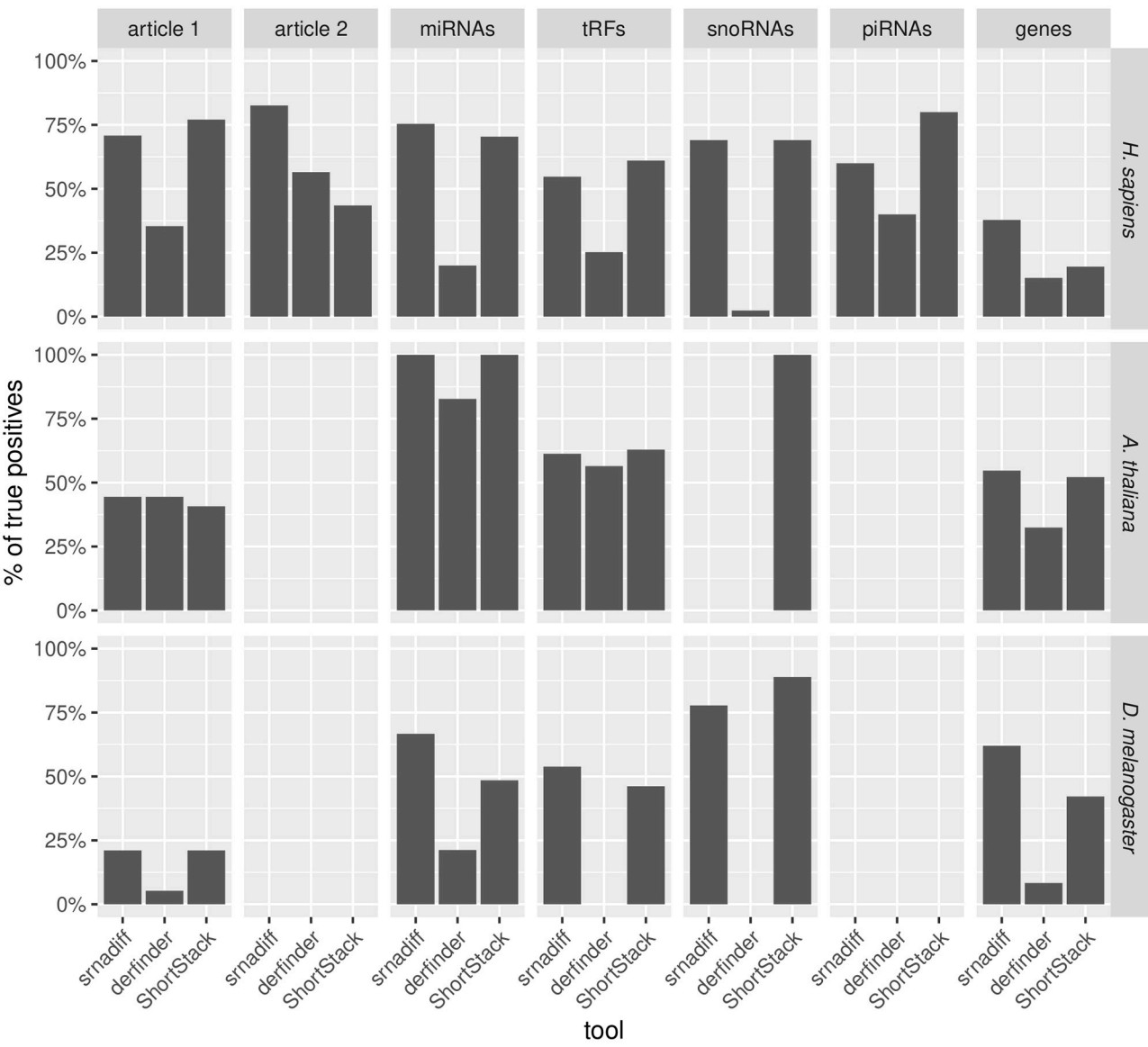

**Fig 4. Recall of the methods compared.** The recall is the percentage of true positives found by the three methods which are evaluated (srnadiff, derfinder, and ShortStack), when compared to a dataset supposed to be *bona fide* elements. The first two columns provide the recall of the methods, when compared to the results published in the papers. The last columns compare the methods to the direct approach, which uses the annotation and performs the test.

**Table 1. Comparison of the results given by the srnadiff, derfinder, and ShortStack methods.** Each cell gives the number of elements found by the tool mentioned in row name that overlap with a region found by the tool mentioned in column name. For instance, 660 regions found by srnadiff overlap with the regions found by Short-Stack. Note that only 93 regions found by ShortStack overlap with regions found by srnadiff. Indeed, ShortStack tend to produce longer regions, and one ShortStack region may overlap several srnadiff regions.

| source | srnadiff | derfinder | ShortStack |
|---|---|---|---|
| srnadiff | — | 21 | 660 |
| derfinder | 27 | — | 26 |
| ShortStack | 93 | 1 | — |

difference point-wise is not significant. The ShortStack method also finds a few more miRNAs found by the articles. However, srnadiff finds, in general, more than thrice as many regions.

Then, we compared all the results with a "truth set" (see Methods and Fig 4), where we retrieved several annotations, performed differential expression, and kept the regions with an adjusted p-value of 5%. We found, for instance, 240 differentially expressed miRNAs, 95 differentially expressed tRFs, etc. srnadiff usually is the method that recovers the greatest number of regions, although ShortStack sometimes provides more. srnadiff found 181 of the 240 differentially expressed miRNAs, 52 of the 97 tRFs, etc. Again, most of the missed features were due to the adjusted p-value threshold effect.

Last, srnadiff discovered 1581 differentially expressed regions outside known small RNA genes, and 809 differentially expressed regions outside known small RNA genes and any Ensembl annotation.

**Biological interpretation.** We wanted then to characterize the differentially expressed regions which do not overlap with known small RNAs. We first checked whether these regions were detected by one, two, or three tools. Fig 5 shows the relative distribution of these regions with respect to protein-coding genes. We first confirmed results shown in Fig 3: the majority of these regions originate from exons of protein-coding genes. These could be degradation products, or silencing RNAs. Some differentially expressed regions are located inside gene promoters, which often also contain some small RNAs [43]. A significant proportion also comes from introns, which are also known to contain regulatory elements, as well miRNAs [44], and snoRNAs [45].

Fig 6 shows the size distribution of these regions. The regions could be attributed to a specific class, depending on their sizes: the 15–20 peak could be putative tRFs, 20–24-nucleotide long regions could be miRNAs, 26–31, piRNAs. Longer regions, often colocalizing with exons or introns, could be degradation products.

We used several additional databases (see Methods) to check whether these unknown differentially expressed regions could be further annotated. We could label two new regions as tRFs, 31 as snoRNAs, and 30 as piRNAs. We used miRDeep2 to discover new putative miRNAs, based on our reads. We could only identify 13 putative new miRNAs this way.

This left 809 unannotated differentially expressed regions, and only 101 if we discard the regions that were located inside or near genes. Two of these regions, uniquely found by srnadiff, are shown in Supporting information. Among these 101 regions, only one was larger than 100 nucleotide, and it does not have the expected profile or a piRNA (widespread distribution of the reads, over-representation of uridines at the first position and adenines at the tenth position. This seems to indicate that the unannotated differentially expressed regions are probably not piRNAs.

## A. thaliana dataset

Results can be found in Figs 3 and 4 and Table 2.

We applied the same methodology as previously. Here, more than half of the miRNAs found in the article was not detected by any other method, mostly because of the p-value threshold effect.

Here again, srnadiff usually gives better results than any other tool, with the exception of the differentially expressed snoRNAs.

## D. melanogaster dataset

Results can be found in Figs 3 and 4, and Table 3.

Similar conclusion can be drawn from this dataset.

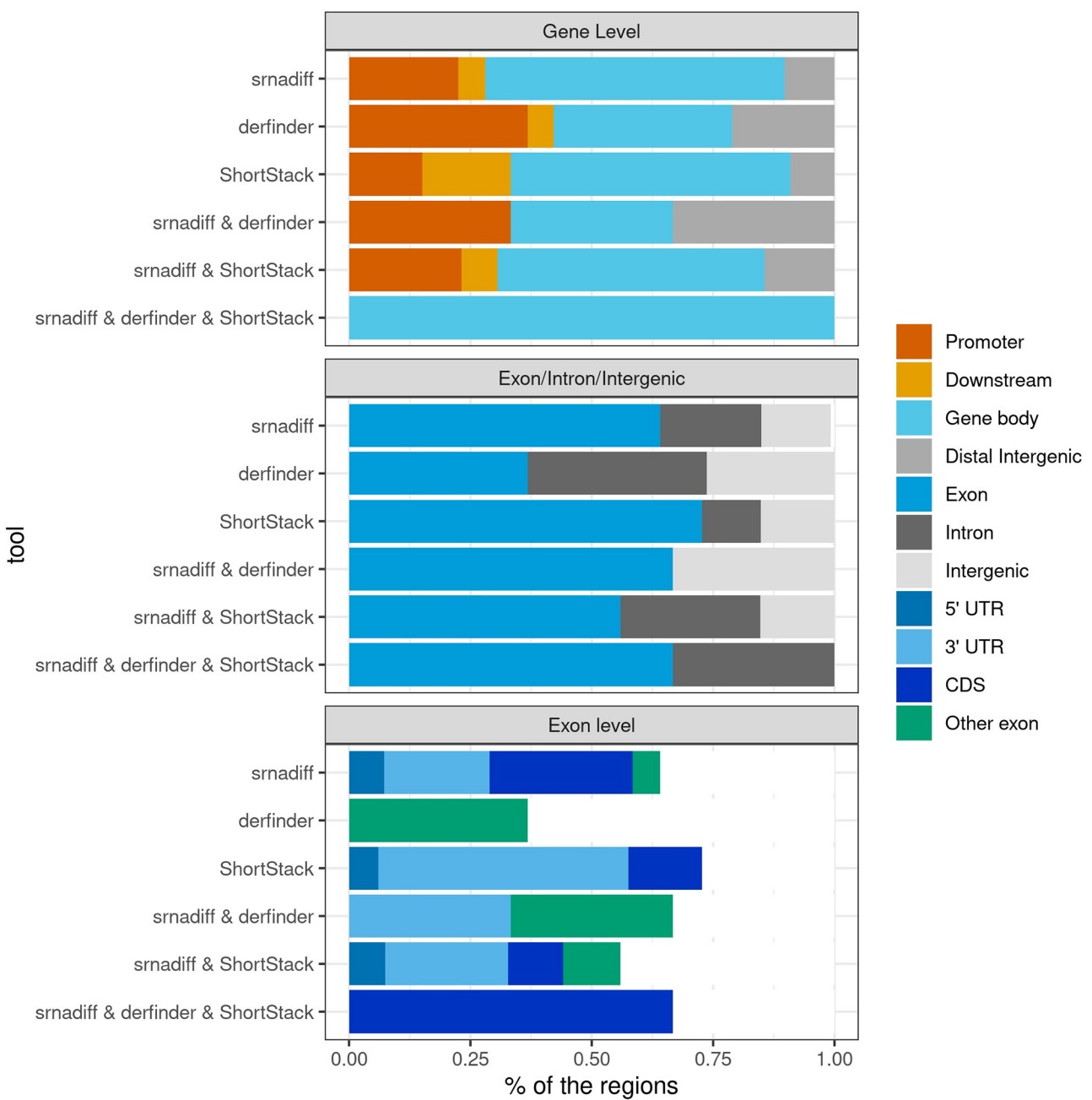

**Fig 5. Genomic distribution of uncharacterized differentially expressed regions.** Each bar plot shows the distribution of the differentially expressed regions outside of known miRNAs, piRNAs, tRFs, snoRNAs, found by one of the tools, or a combination thereof. In the top bar plots, the promoter is defined as the upstream regions of the genes, up to 2000 nucleotides. Likewise, downstream regions are 2000-nucleotide long.

## Synthetic reads

The "truth" set contained 44 differentially expressed features. The results of each tool were plotted as receiver operating characteristic (ROC) curves, given in Fig 7. The ROC curve provides the sensitivity (or recall) against (1 − specificity). The specificity is defined by $\frac{TN}{TN+FP}$, where *TN* is the number of true negatives, and *FP*, the number of false positives. Informally, the area under the curve (AUC) gives the probability that a tool ranks a randomly chosen true

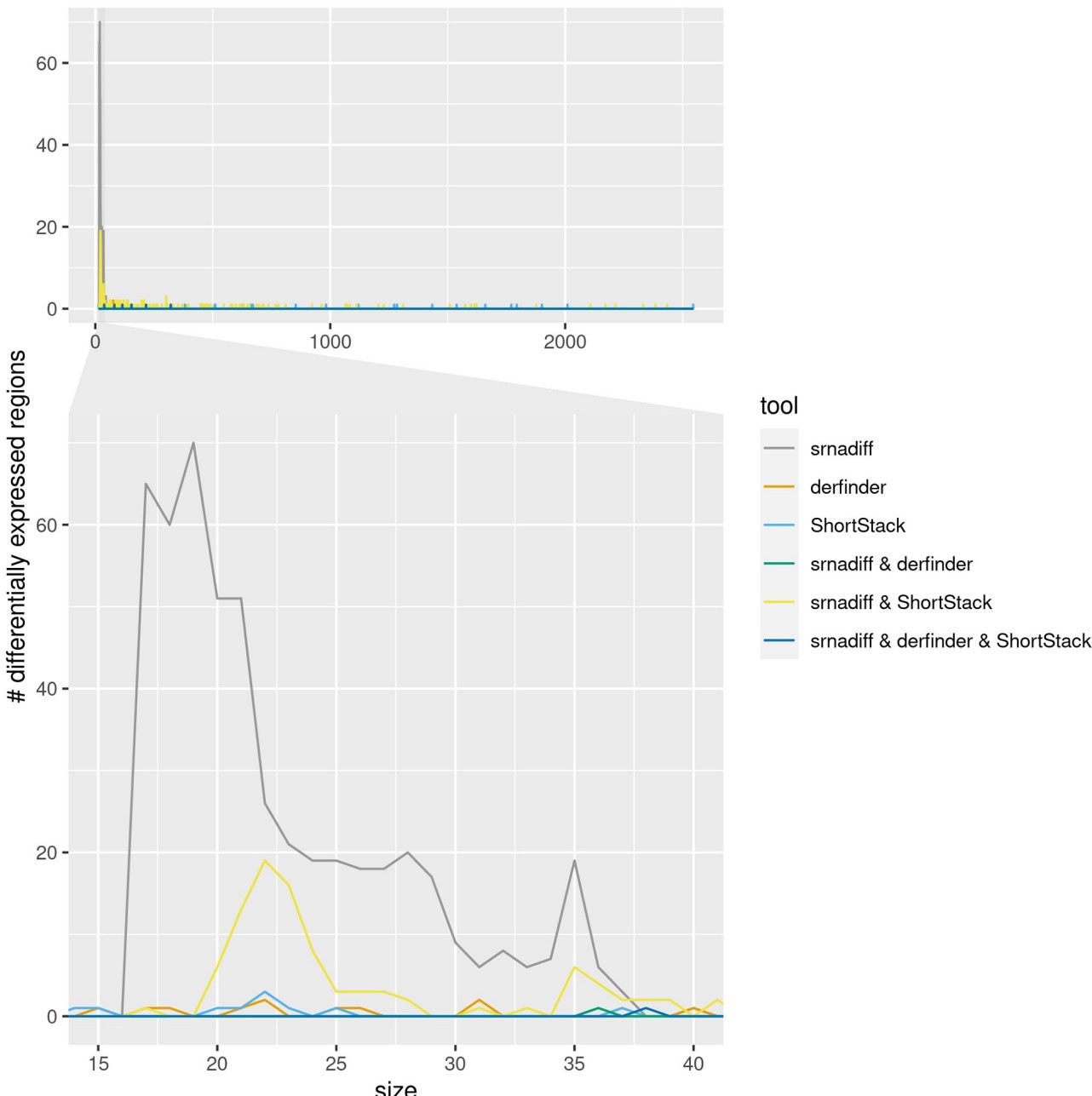

**Fig 6. Size distribution of uncharacterized differentially expressed regions.** The top figure shows the distribution of the size of the differentially expressed regions outside of known miRNAs, piRNAs, tRFs, snoRNAs, found by one of the tools, or a combination thereof. The bottom figure zooms on the 15–40 size distribution.

**Table 2. Comparison of the results given by the srnadiff, derfinder, and ShortStack methods on the *A. thaliana* dataset.** See Table 1 for the meaning of each column.

| method | srnadiff | derfinder | ShortStack |
|---|---|---|---|
| srnadiff | — | 3489 | 7569 |
| derfinder | 4245 | — | 3274 |
| ShortStack | 467 | 112 | — |

**Table 3. Comparison of the results given by the srnadiff, derfinder, and ShortStack methods on the *D. melanogaster* dataset.** See Table 1 for the meaning of each column.

| method | srnadiff | derfinder | ShortStack |
|---|---|---|---|
| srnadiff | — | 164 | 1152 |
| derfinder | 164 | — | 201 |
| ShortStack | 911 | 98 | — |

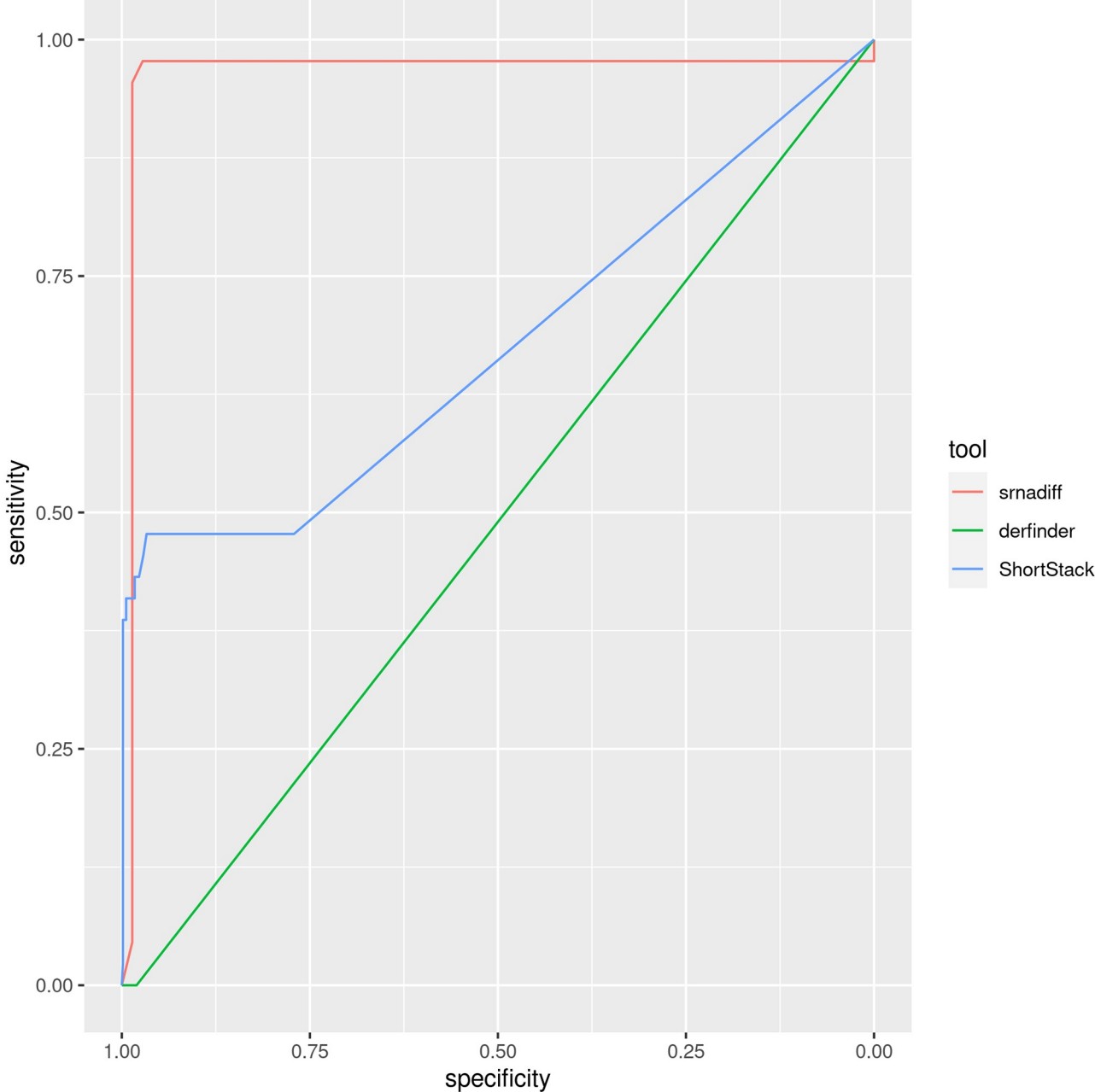

**Fig 7. Results on the synthetic dataset.** The areas under the curves (AUC) of the receiver operating characteristic (ROC) curves of srnadiff, derfinder, and ShortStack are respectively 0.9639, 0.5098, and 0.3235. The (unfiltered) number of candidates for each tool is 72, 60, and 720.

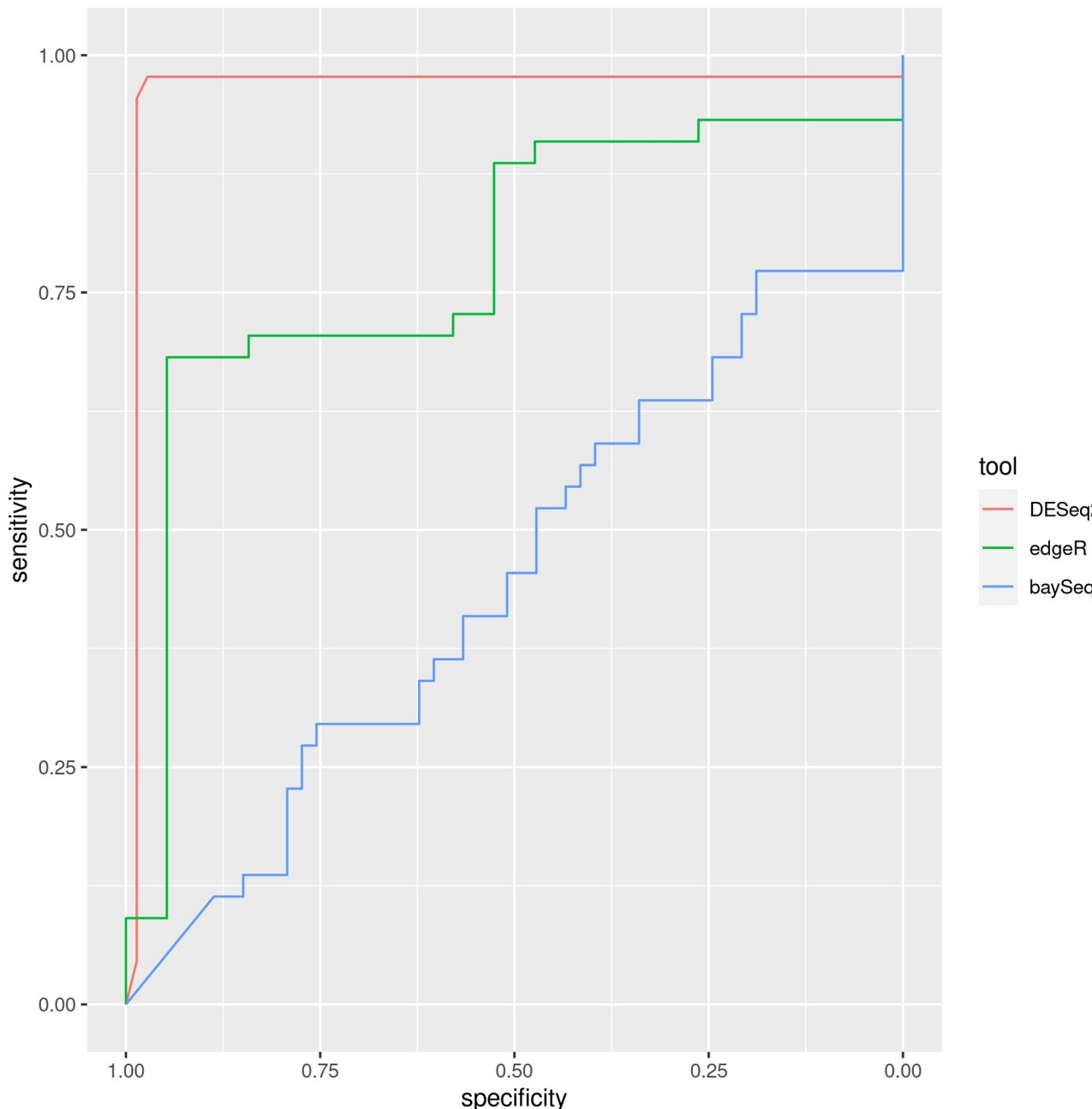

**Fig 8. Effect of the differential expression package on the simulated dataset.** The areas under the curves (AUC) of srnadiff, when using DESeq2, edgeR, or baySeq, is respectively 0.9639, 0.7835 and 0.4468. The (unfiltered) number of candidates for each tool is 72, 19, and 53.

candidate higher than a randomly chosen wrong one. The figure shows that srnadiff clearly outperforms other tools.

We also compared, in this controlled dataset, the effect of the differential analysis package. Fig 8 shows the ROC curve of srnadiff, when using DESeq2 (the default), edgeR, or baySeq. The figure shows that DESeq2 clearly outperforms other packages.

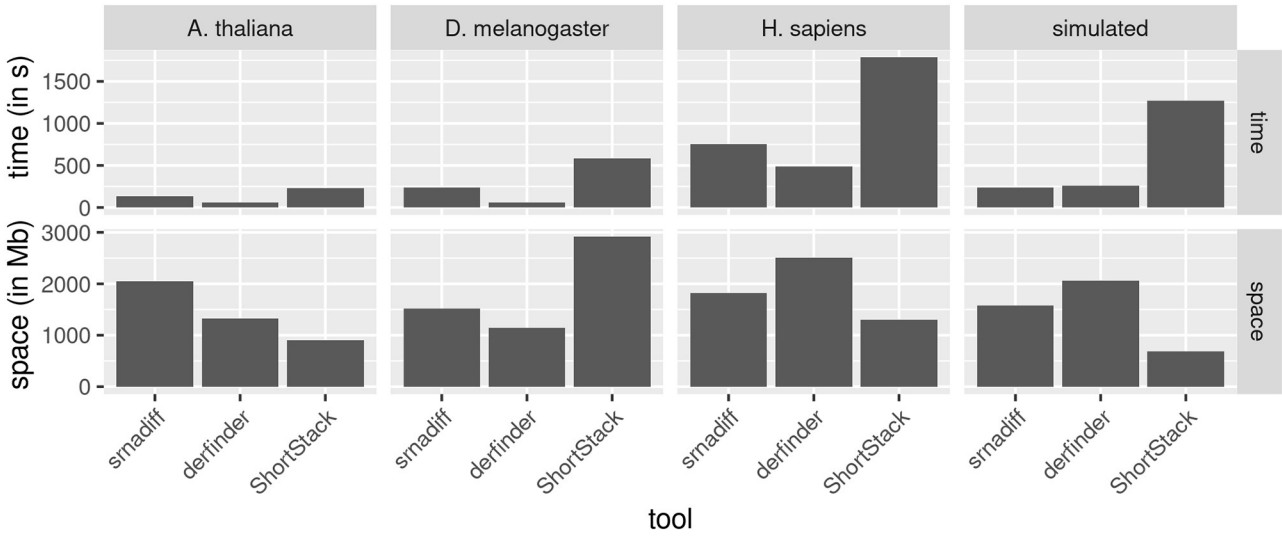

**Fig 9. Time usage of the tools (in seconds), and memory usage (in Mb).**

## Time and memory usage

Fig 9 shows that derfinder is the fastest tool, and ShortStack the slowest. However, srnadiff still provides results within 15 minutes. The reason of the difference between srnadiff and derfinder is that the former implements two methods, and thus processes the data twice. Second, derfinder uses bigWig files, whereas srnadiff readily uses BAM files (and internally converts them into a similar format, which is the bottleneck of the method). ShortStack is a Perl file, which requires significantly more time to process the data.

Concerning the memory usage, derfinder is also the most efficient, and srnadiff usually the least efficient. However, all these computations fit in a standard computer.

All the computation has been performed on a personal computer running Linux Ubuntu 19.04, with Intel Xeon Processor E5–1650 v4 running 6 cores at 3.6 GHz and 32GB RAM.

Benchmarking the preprocessing steps (i.e. conversion from BAM to bigWig, and merging the BAM files for input of ShortStack) is not straightforward, because several pipe-lines are possible. We provide the usage we observed with deepTools [46] to convert BAM files to big-Wig and samtools [47] to merge the BAM files in S1 Appendix.

Other benchmarking, available in S1 Appendix, shows that changing srnadiff parameter does not significantly alter the results. We also show that time increases linearly with the input size. On the other hand, we also showed (see S1 Appendix) that the coverage does not have a dramatic influence on the results.

## Conclusion

In this paper, we propose a new method, called srnadiff, for the detection of differentially expressed small RNAs. The method offers several advantages. First, it can be applied to detect any type of small RNA: miRNAs, tRFs, siRNAs, etc. Second, it does not need any other knowledge on the studied small RNAs, such as a genome annotation, or a set of reference sequences. Moreover, results are comparable to *ad hoc* methods, which detect only a given type of small RNAs.

Our aim is to provide a simple tool that is able to extract all the information given by sRNA-Seq, not only restricting to miRNAs. We hope that srnadiff will make it possible to find new mechanisms involving understudied small RNAs.

Future directions for improvement include broaden the regions found, and providing new strategies to find differentially regions (besides the HMM and the IR methods).

## Supporting information

**S1 Appendix. Additional data.** Supplementary figures, other benchmarking, code used, tool versions, and DOI of the preprocessed data are given in the Additional Data file. (PDF)

## Acknowledgments

The authors thank Hervé Vaucheret, Stéphane Robin and Nathalie Vialaneix for fruitful discussions on the project.

## Author Contributions

**Conceptualization:** Matthias Zytnicki, Ignacio González.

**Investigation:** Matthias Zytnicki.

**Methodology:** Matthias Zytnicki.

**Software:** Matthias Zytnicki, Ignacio González.

**Validation:** Ignacio González.

**Visualization:** Matthias Zytnicki.

**Writing – original draft:** Matthias Zytnicki, Ignacio González.

**Writing – review & editing:** Matthias Zytnicki, Ignacio González.

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
