## [Decision Letter · Decision Letter 0]

19 Jan 2021

PONE-D-20-29322

Finding differentially expressed sRNA-Seq regions with srnadiff

PLOS ONE

Dear Dr. Matthias,

Thank you for submitting your manuscript to PLOS ONE. After careful consideration, we feel that it has merit but does not fully meet PLOS ONE’s publication criteria as it currently stands. Therefore, we invite you to submit a revised version of the manuscript that addresses the points raised during the review process.

We look forward to receiving your revised manuscript.

Kind regards,

J Francis Borgio, Ph.D.,

Academic Editor

PLOS ONE

Journal Requirements:

2.) We note that you have stated that you will provide repository information for your data at acceptance. Should your manuscript be accepted for publication, we will hold it until you provide the relevant accession numbers or DOIs necessary to access your data. If you wish to make changes to your Data Availability statement, please describe these changes in your cover letter and we will update your Data Availability statement to reflect the information you provide.

Additional Editor Comments (if provided):

Through revision is mandatory as per the suggestions by the reviewers. Language style of the MS to be improved by a native speaker.

Reviewers' comments:

Reviewer's Responses to Questions

**Comments to the Author**

1. Is the manuscript technically sound, and do the data support the conclusions?

Reviewer #1: No

Reviewer #2: Yes

Reviewer #3: No

2. Has the statistical analysis been performed appropriately and rigorously? 

Reviewer #1: No

Reviewer #2: Yes

Reviewer #3: Yes

3. Have the authors made all data underlying the findings in their manuscript fully available?

Reviewer #1: No

Reviewer #2: Yes

Reviewer #3: Yes

4. Is the manuscript presented in an intelligible fashion and written in standard English?

Reviewer #1: No

Reviewer #2: Yes

Reviewer #3: No

5. Review Comments to the Author

Reviewer #1: I have tried to read the manuscript carefully a couple of times to get the idea of it and especially the novelty as was claimed. Unfortunately it was so hard for me to understand it. The manuscript seems to me that it was prepared in rush and not in a careful manner. This seems not a research study rather than just a suggestion for solving a problem of simple question.

Reviewer #2: This manuscript describes an R package, 'srnadiff', which is designed to a) identify genomic clusters of small RNA coverage, and b) use the count data from the clusters to run a differential expression analysis. The tool uses the widely used DESeq2 package to run differential expression analysis; thus the major innovations are the small RNA locus identifications, and the automated nature of passing everything directly to an established differential expression analysis method. The package also comes with a useful plotting tool for visualizing small RNA coverage at discovered loci.

The manuscript is generally well-organized and contains appropriate benchmarking analyses. The package is available on Bioconductor, and there is ample documentation. I was able to install the package and run the example data on my laptop with minimal issues. I was also able to analyze my own 'real' data with minimal problems.

Overall this is a useful and well-documented tool that should find use in the community. The manuscript clearly meets the criteria for publication in PLOS One. I have some minimal comments, in two categories: Things that should be done before publication, and general suggestions / comments / fixes for the authors to consider, mostly related to the documentation / vignette.

Suggested changes prior to publication:

1. Abstract: "micro RNA" should be "microRNA" (all one word).

2. Figure 5 caption: "computer" should be "computed".

3. Table 1 was very confusing to me. I read the caption several times but I still did not really understand what each number means. Seems like two separate items may be conflated (locus annotation and diff exp. testing?). Anyway please re-think this table. It may seem simple to you, but I could not adequately understand it.

4. Same comment as above for Table 2 and Table 3.

5. Table 5 I think should also be modified. Currently the columns aren't really labelled correctly. Having two numbers, in different units, in the same cell is very confusing.

Other comments / documentation suggestions / bugfixes to consider

6. In the vignette: there is a duplicated code line and comment; the first instance is in the wrong location (can't be run yet because variable 'sampleInfo' not yet set):

## Vector with the full paths to the BAM files to use

bamFiles <- paste(file.path(basedir, sampleInfo$FileName), "bam", sep=".")

7. Maybe it's just me but I find GenomicRanges objects confusing and a little annoying. I suggest modifying the vignette to give users more guidance on how to extract all discovered regions into a commonly used format (gff3, bed, or just a tsv flatfile)? Put in vignette at least the 'as.data.frame()' idiom. Maybe people who are already used to the Bioconductor / GenomicRanges world are used to it, but others (like me) aren't, and at the end of the day I think most people need a flat file.

8. Vignette 5.1 : "The output in a GenomicRanges object, and the information is accessible with the mcols() function" ... no mcols() function is loaded. Again outputting as a simple table would be most useful.

9. Some praise: plotRegions() function is very nice.

10. Do the BAM files need to sorted and indexed? It seems like they do. Should specify in the vignette / documentation that if not indexed, they will be automatically by the package, and that they should be coordinate sorted first.

11. sampleInfo : required column 'FileName' is not actually the file name (.bam is stripped off). This may be confusing to users. Seems like it should just be the file's actual name, including the .bam.

12. Directionality: log2FoldChange, which condition is denominator, which is numerator? It is very hard (impossible?) to tell. A more general comment here is that perhaps the user should be allowed to more directly interface with DESeq2?

13. When I ran my real world dataset (not the toy dataset provided with the package), plotRegions failed with cryptic error:

> plotRegions(srnaExp, regions(srnaExp)[1])

Error in .fillWithDefaults(data.frame(start = as.integer(start), end = as.integer(end)), :

Number of elements in argument 'id' is invalid

Reviewer #3: The manuscript by Zytnicki and Gonzalez presents a new method to identify small RNAs (sRNA) differentially expressed, going beyond the classic miRNAs and most known sRNAs that can be mapped to well defined precursors.

In principle the tool can be useful. The introduction reasonably well presents the state of the art in the field, with some important elements lacking. Unfortunately, the result presentation must be reorganized, improving figures, clarity and language.

I have some suggestions to improve the paper.

Major:

1. Long RNA are >200 nt. This does not mean that sRNAs are <200 nt. Instead normally sRNAs are <50 nt, with 50-200 being a grey zone of average sized RNAs difficult to study with classic protocols for long or short RNA-seq.

2. In the Introduction, when approaches to “go beyond miRNAs” are cited (citation 5), please discuss and cite also: MiR&moRe2: A Bioinformatics Tool to Characterize microRNAs and microRNA-Offset RNAs from Small RNA-Seq Data. Int J Mol Sci. 2020

3. From very similar studies this reviewer knows well that RNA degradation, specific experimental conditions and protocols can mimic sRNA expression while longer RNAs are actually present. The manuscript should better consider this point. Please see and cite: RNY4 in Circulating Exosomes of Patients With Pediatric Anaplastic Large Cell Lymphoma: An Active Player? Front Oncol. 2020 in which sRNAs are identified with a method similar to srnadiff and then validations disclosed the the entire RNAY4 was present. See also Driedonks and Nolte-'t Hoen Driedonks TAP, Nolte-'t Hoen ENM. Front Immunol. (2018), a study that suggested that YRNA secondary structures might impede full-length cDNA synthesis, leading to overestimation of fragmented non-coding RNA (sRNA) in sequencing data.

4. It is not clear to me why the so called ShortStack method is not described in the Introduction section when Derfinder is mentioned, since both were compared with srnadiff.

5. The Figures are strangely separated, each composed by a simple panel. Consider merging Figures 2-7 into one or two better designed figures, thematically.

6. The result presentation is far too preliminary. Results are too schematic and should be presented ad commented in a different way. F.i. figures with overlap of methods results, considering different factors (length of regions, type of annotation, etc.) can be useful and are completely missing.

7. Several sentences must be revised for language or because they are incorrect or not meaningful from a biological point of view.

- “have been identified as a key actor to study and understand the development of the cell.” It’s a key actor in determing cell behavior, not for the study of…

- “and have an imprecise “gene” structure”. Please change, saying things how they are: not well characterized precursor RNA

- “The role of these sRNAs is usually understood via a differential expression protocol, 9 e.g. healthy vs sick, or wild type vs mutant. “ the comparison aims to identify expression dysregulation, not the role of sRNAs

6. PLOS authors have the option to publish the peer review history of their article (what does this mean?). If published, this will include your full peer review and any attached files.

Reviewer #1: No

Reviewer #2: **Yes: **Michael J. Axtell

Reviewer #3: No

---

## [Author Response · Author response to Decision Letter 0]

13 Feb 2021

Dear Editor, Dear Reviewers,

First, we would like to thank Reviewers #2 and #3 for their analysis, and useful suggestions. We tried to alter the manuscript, the vignette, and to code, to meet the requirements. We hope that you will find our work useful for the community.

Here are our answers to the Reviews.

Best regards,

Ignacio and Matthias.

Reviewer #1: I have tried to read the manuscript carefully a couple of times to get the idea of it and especially the novelty as was claimed. Unfortunately it was so hard for me to understand it. The manuscript seems to me that it was prepared in rush and not in a careful manner. This seems not a research study rather than just a suggestion for solving a problem of simple question.

==> We respectfully disagree here. The problem may not be well explained in this paper, but finding differentially expressed small RNAs without an annotation is not a "problem of simple question." The "derfinder2" paper clearly presents the difficulty of the problem, in the mRNA context. However, if you have an obvious answer, we would be more than happy if you could share it with the community, who is eagerly waiting for such a method, especially for researchers who work on species where the small RNAs are poorly characterized.

Reviewer #2: This manuscript describes an R package, 'srnadiff', which is designed to a) identify genomic clusters of small RNA coverage, and b) use the count data from the clusters to run a differential expression analysis. The tool uses the widely used DESeq2 package to run differential expression analysis; thus the major innovations are the small RNA locus identifications, and the automated nature of passing everything directly to an established differential expression analysis method. The package also comes with a useful plotting tool for visualizing small RNA coverage at discovered loci.

The manuscript is generally well-organized and contains appropriate benchmarking analyses. The package is available on Bioconductor, and there is ample documentation. I was able to install the package and run the example data on my laptop with minimal issues. I was also able to analyze my own 'real' data with minimal problems.

Overall this is a useful and well-documented tool that should find use in the community. The manuscript clearly meets the criteria for publication in PLOS One. I have some minimal comments, in two categories: Things that should be done before publication, and general suggestions / comments / fixes for the authors to consider, mostly related to the documentation / vignette.

Suggested changes prior to publication:

1. Abstract: "micro RNA" should be "microRNA" (all one word).

==> Done.

2. Figure 5 caption: "computer" should be "computed".

==> Done.

3. Table 1 was very confusing to me. I read the caption several times but I still did not really understand what each number means. Seems like two separate items may be conflated (locus annotation and diff exp. testing?). Anyway please re-think this table. It may seem simple to you, but I could not adequately understand it.

==> Reviewer 3 also had a comment on the tables, suggesting a transformation in figures. Part of the tables are now figures, hopefully easier to read. The remaining table are more carefully explained:

===

Comparison of the results given by the srnadiff, derfinder, and ShortStack methods.

Each cell gives the number of elements found by the tool mentioned in row name that overlap with a region found by the tool mentioned in column name. For instance, 579 regions found by srnadiff overlap with the regions found by ShortStack. Note that only 463 regions found by ShortStack overlap with regions found by srnadiff. Indeed, ShortStack tend to produce longer regions, and one ShortStack region may overlap several srnadiff regions.

4. Same comment as above for Table 2 and Table 3.

==> We did the same.

5. Table 5 I think should also be modified. Currently the columns aren't really labelled correctly. Having two numbers, in different units, in the same cell is very confusing.

==> Following suggestion by reviewer 3, we transformed the tables to 2 figures.

Other comments / documentation suggestions / bugfixes to consider

6. In the vignette: there is a duplicated code line and comment; the first instance is in the wrong location (can't be run yet because variable 'sampleInfo' not yet set):

## Vector with the full paths to the BAM files to use

bamFiles <- paste(file.path(basedir, sampleInfo$FileName), "bam", sep=".")

==> We corrected it, thanks!

7. Maybe it's just me but I find GenomicRanges objects confusing and a little annoying. I suggest modifying the vignette to give users more guidance on how to extract all discovered regions into a commonly used format (gff3, bed, or just a tsv flatfile)? Put in vignette at least the 'as.data.frame()' idiom. Maybe people who are already used to the Bioconductor / GenomicRanges world are used to it, but others (like me) aren't, and at the end of the day I think most people need a flat file.

==> Fortunately, it is very easy to export to a flat file.

I added this in the vignette:

===

You can export the regions to a BED file with the "rtracklayer" function `export`.

```

library(rtracklayer)

export(regions, "file.bed")

```

You can either export to GFF, GFF3, or GTF formats. Simply change the file suffix.

===

We also added a similar comment in the paper:

===

The "regions" function provides the differentially expressed regions, in a "GenomicRanges" object, which can be exported to BED, GTF, or GFF files with the rtracklayer package.

===

8. Vignette 5.1 : "The output in a GenomicRanges object, and the information is accessible with the mcols() function" ... no mcols() function is loaded. Again outputting as a simple table would be most useful.

==> Strange, I can use the mcols function. Yet the problem has been solved!

9. Some praise: plotRegions() function is very nice.

==> Thank you very much!

10. Do the BAM files need to sorted and indexed? It seems like they do. Should specify in the vignette / documentation that if not indexed, they will be automatically by the package, and that they should be coordinate sorted first.

==> As you guessed, the BAM files should be sorted beforehand. You also noticed that the index is created, if missing. We added this in the vignette:

 - Quick start: The user needs to provide a vector with the full paths to the BAM files (which should be coordinate sorted),

 - Data preparation: **paths to the BAM files:** a vector with the full paths to the sample BAM files, which should be coordinate sorted. Please use the `sortBam` function of the "Rsamtools" package to sort the reads.

11. sampleInfo : required column 'FileName' is not actually the file name (.bam is stripped off). This may be confusing to users. Seems like it should just be the file's actual name, including the .bam.

==> Very true. We changed the code accordingly. The package now needs the file suffix (".bam"). We also updated the vignette.

12. Directionality: log2FoldChange, which condition is denominator, which is numerator? It is very hard (impossible?) to tell. A more general comment here is that perhaps the user should be allowed to more directly interface with DESeq2?

==> This is true that directionality is not easy to predict. We added a new section in the vignette:

===

### Directionality

`srnadiff` chooses a reference condition, which is the control, or wild type condition. The results reflect the comparison of the other condition versus the reference condition. By default, the reference condition is the first condition, when the conditions are sorted by the alphabetical order. It is thus advisable to specify the reference condition.

As mentionned in `DESeq2`, you can select the reference condition using factors:

```

sampleInfo$Condition <- factor(sampleInfo$Condition, levels = c("control", "infected"))

```

Internally, `srnadiff` (through `DESeq2\\) computes the log-fold-change of the comparison. The log-fold-change is positive when the reference condition is less expressed than the other condition.

===

Interacting with DESeq2 is tricky. Actually, we use it twice: once in the HMM step, and one in the final step. And after the DESeq2 step, there is still some post-processing (because non-optimal regions are discarded).

13. When I ran my real world dataset (not the toy dataset provided with the package), plotRegions failed with cryptic error:

> plotRegions(srnaExp, regions(srnaExp)[1])

Error in .fillWithDefaults(data.frame(start = as.integer(start), end = as.integer(end)), :

Number of elements in argument 'id' is invalid

==> I am sorry that we cannot reproduce your error.

We also used our package on at least the 4 datasets which are evaluated in this paper, and we do not see this error.

Could it be possible for you to share you data, and post an Issue to GitHub with the code you used?

Reviewer #3: The manuscript by Zytnicki and Gonzalez presents a new method to identify small RNAs (sRNA) differentially expressed, going beyond the classic miRNAs and most known sRNAs that can be mapped to well defined precursors.

In principle the tool can be useful. The introduction reasonably well presents the state of the art in the field, with some important elements lacking. Unfortunately, the result presentation must be reorganized, improving figures, clarity and language.

I have some suggestions to improve the paper.

Major:

1. Long RNA are >200 nt. This does not mean that sRNAs are <200 nt. Instead normally sRNAs are <50 nt, with 50-200 being a grey zone of average sized RNAs difficult to study with classic protocols for long or short RNA-seq.

==> Thank you very much for the comment. I do notice, however, that some well-established publication also use the 200 bp definition for sRNAs:

 - Post-transcriptional processing generates a diversity of 5'-modified long and short RNAs. Nature. 2009.

 - A mouse tissue atlas of small noncoding RNA. PNAS. 2020.

Moreover, I would not say that sRNAs "normally" are <50bp, as this would exclude most snoRNAs, for instance.

In my personal experience, I did not find it very difficult to study small RNAs up to 200bp.

Yet, I modified my text accordingly:

===

The majority of these RNA contain less than 50 bp, but some may reach up to 200 bp.

===

2. In the Introduction, when approaches to “go beyond miRNAs” are cited (citation 5), please discuss and cite also: MiR&moRe2: A Bioinformatics Tool to Characterize microRNAs and microRNA-Offset RNAs from Small RNA-Seq Data. Int J Mol Sci. 2020

==> Thank you very much for the suggestion.

We added a short description of several tools that can be used to detect specific small RNAs.

However, we are not aware that MiR&moRe2 is widely used in the community to annotate small RNAs, and, because of space constraints, restricted our description to the most widely used tools.

We added this:

===

It is, at least in principle, always possible to complement the annotation of the genome of interest using tools that detect a class of small RNAs \\textit{de novo}: tRNAscan-SE for tRNAs, Snoscan for snoRNAs, or infeRNAl for small RNAs described in the RFAM database are the most widely used tools.

However, this requires expertise, as well as computational resources.

It is, in practice, rarely done, and still restricts the analysis to well-characterized small RNAs.

===

3. From very similar studies this reviewer knows well that RNA degradation, specific experimental conditions and protocols can mimic sRNA expression while longer RNAs are actually present. The manuscript should better consider this point. Please see and cite: RNY4 in Circulating Exosomes of Patients With Pediatric Anaplastic Large Cell Lymphoma: An Active Player? Front Oncol. 2020 in which sRNAs are identified with a method similar to srnadiff and then validations disclosed the the entire RNAY4 was present. See also Driedonks and Nolte-'t Hoen Driedonks TAP, Nolte-'t Hoen ENM. Front Immunol. (2018), a study that suggested that YRNA secondary structures might impede full-length cDNA synthesis, leading to overestimation of fragmented non-coding RNA (sRNA) in sequencing data.

==> Thank you very much for this comment.

It is true that small RNA sequencing has some biases. However, we preferred citing the following article, which makes an in-depth analysis of the possible biases, which include nucleotide content at the extremities, GC%, and secondary structures, amoung other:

Bias in Ligation-Based Small RNA Sequencing Library Construction Is Determined by Adaptor and RNA Structure

Fuchs RT, Sun Z, Zhuang F, Robb GB (2015) Bias in Ligation-Based Small RNA Sequencing Library Construction Is Determined by Adaptor and RNA Structure. PLOS ONE 10(5): e0126049. https://doi.org/10.1371/journal.pone.0126049

So, we added the following paragraph in the Introduction:

===

Sequencing these sRNAs is now the state-of-art technique in order to study them.

It accurately quantifies all the sRNAs in a single experiments --known and unknown sRNAs alike-- even though it suffers from some known biases.

===

4. It is not clear to me why the so called ShortStack method is not described in the Introduction section when Derfinder is mentioned, since both were compared with srnadiff.

==> We did mention and briefly present ShortStack in the Introduction, slightly before derfinder.

However, I did not mention it in the Author Summary, you probably wanted to refer to this section! In this case, we did add it there, thanks!

===

Since our method is the first one to use the \\textit{identify-then-annotate} strategy on sRNAs, we compared our method against a similar method, developed for long RNAs (derfinder), and to the annotate-then-identify strategy, where the sRNAs have been identified beforehand using a segmentation tool (here, ShortStack), on three published datasets, and a simulated one.

5. The Figures are strangely separated, each composed by a simple panel. Consider merging Figures 2-7 into one or two better designed figures, thematically.

==> Certainly. We gathered figures 2 to 7 in one, single figure.

6. The result presentation is far too preliminary. Results are too schematic and should be presented ad commented in a different way. F.i. figures with overlap of methods results, considering different factors (length of regions, type of annotation, etc.) can be useful and are completely missing.

==> As you suggested, we transformed some tables to figures. There are still some tables left, because we cannot think of a simple and visual way to render a table which is not symmetric. For instance, in Table 1, 256 regions found by derfinder overlap with regions found by srnadiff, but only 255 regions found by srnadiff overlap with regions found by derfinder. Indeed, one (longer) regions found by srnadiff overlaps two (shorter) regions found by derfinder. Venn diagrams, or upset plots do not accommodate with this type of dataset.

Note that benchmarking on the type of annotation (miRNAs, tRFs, snoRNAs, piRNAs, and genes) was present in the table (admittedly not very readable). As you suggested, the results are now in a figure, here Figure 4.

The lengths of the regions was also compared, in Supporting Information (Figures 1 and 2 for the H. sapiens dataset). We also compared the fact that some tools split differentially expressed regions into smaller ones (Figure 3), and the p-value distributions (Figure 4).

7. Several sentences must be revised for language or because they are incorrect or not meaningful from a biological point of view.

- “have been identified as a key actor to study and understand the development of the cell.” It’s a key actor in determing cell behavior, not for the study of…

- “and have an imprecise “gene” structure”. Please change, saying things how they are: not well characterized precursor RNA

- “The role of these sRNAs is usually understood via a differential expression protocol, 9 e.g. healthy vs sick, or wild type vs mutant. “ the comparison aims to identify expression dysregulation, not the role of sRNAs

==> Thank you! The sentences should be more correct now:

 - These sRNA have a wide range of activities, which include gene regulation, protection against virus, transposable element silencing, and have been identified as a key actor in determining the development of the cell.

 - However, tools are lacking to detect other types of sRNAs, which are less studied, \\hl{and whose precursor RNA is not well characterized}.

 - The regulation of the expression induced by these sRNAs is usually understood via a differential expression protocol, e.g. healthy vs sick, or wild type vs mutant.

---

## [Decision Letter · Decision Letter 1]

15 Apr 2021

PONE-D-20-29322R1

Finding differentially expressed sRNA-Seq regions with srnadiff

PLOS ONE

Dear Dr. Matthias,

Thank you for submitting your manuscript to PLOS ONE. After careful consideration, we feel that it has merit but does not fully meet PLOS ONE’s publication criteria as it currently stands. Therefore, we invite you to submit a revised version of the manuscript that addresses the points raised during the review process.

We look forward to receiving your revised manuscript.

Kind regards,

J Francis Borgio, Ph.D.,

Academic Editor

PLOS ONE

Reviewers' comments:

Reviewer's Responses to Questions

**Comments to the Author**

1. If the authors have adequately addressed your comments raised in a previous round of review and you feel that this manuscript is now acceptable for publication, you may indicate that here to bypass the “Comments to the Author” section, enter your conflict of interest statement in the “Confidential to Editor” section, and submit your "Accept" recommendation.

Reviewer #2: All comments have been addressed

Reviewer #4: (No Response)

Reviewer #5: (No Response)

2. Is the manuscript technically sound, and do the data support the conclusions?

Reviewer #2: (No Response)

Reviewer #4: Partly

Reviewer #5: Partly

3. Has the statistical analysis been performed appropriately and rigorously? 

Reviewer #2: (No Response)

Reviewer #4: I Don't Know

Reviewer #5: Yes

4. Have the authors made all data underlying the findings in their manuscript fully available?

Reviewer #2: (No Response)

Reviewer #4: Yes

Reviewer #5: No

5. Is the manuscript presented in an intelligible fashion and written in standard English?

Reviewer #2: (No Response)

Reviewer #4: Yes

Reviewer #5: Yes

6. Review Comments to the Author

Reviewer #2: (No Response)

Reviewer #4: The manuscript by Zytnicki and Gonzales describes srnadiff, a new computational method for calling differentially expressed small RNAs de novo, without relying on genome annotation. This direction is biologically attractive, especially given a variety of cell type-specific small RNAs that can be missed in a basic annotation, including, but not limited to, new miRNAs, piRNAs, tRNA fragments, and possibly enhancer RNAs (eRNAs). The implementation of this method is based on DESeq–based analysis of differential expression supplemented by a few straightforward computational steps. The description of the method itself is mostly sufficient, although I have some questions about the choices of methodology. The description of results produced by this method and their evaluation needs more serious attention. For a potential user, it would be important to know what new additional information this method can provide as compared to both the basic method based on pre-existing annotation and to other comparable methods.

Major comments:

1. It looks like positional resolution of this method is 1 bp as suggested by the use of run-length encoding. The standard statistical null model of DESeq, however, is optimized towards the analysis of read counts over whole transcripts. The CPMs/read counts calculated at 1-bp resolution across the genome most likely follow a different random statistical distribution compared to the set of full-length transcripts. Was this assessed?

2. Line 121-122:

If the method’s performance does not depend on the values of HMM parameters, is HMM the optimal computational framework to use?

This lack of dependency may suggest a problem with the assumption of the process based on nucleotide-to-nucleotide transition, a problem with data sensitivity (low depth) etc.

3. In the evaluation section, precision and recall are shown only as barplots. Would it make sense to present curves based on the ranked lists of predictions?

4. Using previously annotated sRNAs as gold standard ("truth set") may be appropriate as a first-level evaluation, but it is not clear how much srnadiff adds to the basic analysis of pre-annotated sRNAs.

It would be important to address the predictions of differentially expressed sRNAs outside of basic annotations, overlaps of these predictions between the three tested methods; and to discuss possible biological relevance and examples of these predictions: for example, potential new miRNAs or piRNAs (hopefully confirmed by the presence of typical sequence patterns), tRNA fragments etc.

5. Lines 292-294: For a potential user, it would be important to know more about the new regions predicted by srnadiff but not by the ShortStack method. How many of these regions are previously annotated and how many are novel? How these novel regions are distributed between the known functional genomic elements according to a standard annotation of the genome: introns, exons, enhancers etc? What is the length distribution of these novel regions?

It would also be important to show examples of newly predicted regions of potential biological interest, similar to the previous comment.

Minor comments:

1. Although the goal is to precisely define sRNA positioning in the genome, the 1 bp resolution of read counts may result in lower read counts and lower sensitivity of differential calls by DESeq. In a similar vein, what is a reasonable threshold of p-value (line 126) at which a genomic position is omitted from statistical consideration? Was it estimated in a rigorous way based on real data?

2. English needs minor corrections and stylistic editing.

Reviewer #5: The work proposed a method called “srnadiff” in order to find differentially expressed sRNA, without annotation (the annotation is optional).

The authors divide the proposed approach into two steps. The first step comprises applying some methods like HMM (hidden Markov model), IR (Irreducible regions), annotation (as optional), etc. in order to produce genomic intervals that are potential differentially expressed regions.

The second step comprises clustering the samples and applying the DESeq2 method for the identification of differential expression.

Essentially, the problem addressed by the paper is not one of detecting differential expression, in numerical or statistical terms, but rather one of mapping, i.e., the contribution of the work is essentially in step 1.

Therefore, the manuscript presents an R package for the detection of differentially expressed smallRNAs. The text is difficult to understand and does not present clearly with tables and figures (see specific comments below). The authors do not sustain the reasons their results are better (identifying more differentially expressed sequences does not exactly mean that the tool is better), perhaps this difficulty in making clear is because of the way the comparison was done.

The work is interesting and applied in a significant bioinformatics research context. Some points could be better presented which could improve the quality of the work.

Major:

Regarding the presentation of the method, it would be very important to improve the presentation of step 1 by clearly contextualizing how the different adopted approaches are integrated to perform the clustering of genomic regions. Making clear the adopted criteria, information flow and parameters considered, among other information.

Explain and include references to which methods and implementations were adopted for the HMM and IR.

The acronym snoRNAs is used but not previously presented as piRNAs for example.

The method describes expression identification with only RNA-Seq data without reference, but in Figure 1 the first step uses a BAM (output file mapping reads to a reference).

In step 2, the proposed work applies the DESeq2 method as a method for identifying differential expression. The proposed approach is available from a Bioconductor package. The suggestion to the authors is to make available to use it as a library and apply other methods for differential expression detection. An interesting proposal would be to apply the methods together, as proposed in this work:

https://journals.plos.org/plosone/article?id=10.1371/journal.pone.0190152

The tool itself is well done and seems to me to be really necessary, even for computational efficiency, but the way it is presented can be much improved, despite the corrections already stated in revision 1.

Regarding the results, in the presentation in Fig. some works are mentioned (article 1, article 2, usual differential expression calling method). It is important to contextualize these works in the text, justify the choice of these papers and their characteristics so that it is possible to compare them contextually with the proposed approach.

Fig. 4 can be better presented and contextualized. It is important that the authors make clear the numbers and definitions of what they considered as True Positive and True Negative for each of the datasets. In addition, it is important that they present the Precision (TP/(TP+FP)) and Recall (TP/(TP+FN)) rates in order to make clear the improvement provided by the proposed method. Clarify that the method recovers regions with differential expression and with precision, i.e., without an excessive number of false positives. Also, correct the y-axis labeling of the figures. It would interest to make clear what these results show. It is important to explain and contextualize where this better result can be visualized and why it is described as “better”.

Fig. 3 refers to the comparison tools as source, 4 tool, 5 and 6 method.

Fig. 6, Y-axis should be better identified what represents “value”?

The p-value was defined for all methods, however the fold-change is mentioned only for synthetic dataset. It is important to make clear the adopted p-value and fold-change for all methods and datasets.

The preprocessed dataset used to generate the results was not made available by the authors. It is recommended that the dataset used be made available that allows the replication of the results by the research community.

Minor:

- Fig. 2 low resolution.

7. PLOS authors have the option to publish the peer review history of their article (what does this mean?). If published, this will include your full peer review and any attached files.

Reviewer #2: **Yes: **Michael J. Axtell

Reviewer #4: No

Reviewer #5: No

---

## [Author Response · Author response to Decision Letter 1]

30 May 2021

Dear Editor,

First, we thank you very much for this thorough review.

We tried to answer all the suggestions and comments addressed by the reviewers, in the paper, as well as in the package.

Please find hereafter our detailed answers.

Best regards,

Ignacio Gonzalez and Matthias Zytnicki.

Reviewer #2: (No Response)

Reviewer #4: The manuscript by Zytnicki and Gonzales describes srnadiff, a new computational method for calling differentially expressed small RNAs de novo, without relying on genome annotation. This direction is biologically attractive, especially given a variety of cell type-specific small RNAs that can be missed in a basic annotation, including, but not limited to, new miRNAs, piRNAs, tRNA fragments, and possibly enhancer RNAs (eRNAs). The implementation of this method is based on DESeq–based analysis of differential expression supplemented by a few straightforward computational steps. The description of the method itself is mostly sufficient, although I have some questions about the choices of methodology. The description of results produced by this method and their evaluation needs more serious attention. For a potential user, it would be important to know what new additional information this method can provide as compared to both the basic method based on pre-existing annotation and to other comparable methods.

Major comments:

1. It looks like positional resolution of this method is 1 bp as suggested by the use of run-length encoding. The standard statistical null model of DESeq, however, is optimized towards the analysis of read counts over whole transcripts. The CPMs/read counts calculated at 1-bp resolution across the genome most likely follow a different random statistical distribution compared to the set of full-length transcripts. Was this assessed?

 ==> We used DESeq2 twice in the pipe-line. First, as a preprocessing step of the HMM method, and second, in order to give a p-value of the candidates in the last step.

 In the latter context, we do not use the run-length encoding version, but we do in the former context. As you commented, it is true that DESeq2 has not been designed for that, and we did not provide the reader enough material to make sure that this use was in line with DESeq2 statistical assumptions.

 We added this in the article:

===

This use of DESeq2 is obviously not standard, because DESeq2 expects the total number of reads per transcript, and not the nucleotide-wise number of reads (also called \\emph{coverage}).

However, we can first notice here is that, in an ideal case of a miRNA, piRNA, or snoRNA, where all the reads do map at the very same positions and cover the full transcript, the total number of reads and the coverage are exactly identical.

We also plotted several quality control distribution, and checked that they behave as expected.

First, we plotted the dispersion estimation.

Briefly, the read counts for a particular gene are expected to follow a negative binomial distribution, with two parameters: the mean, and the dispersion.

The dispersion is estimated using the mean, and is expected to follow a trend for the majority of the genes.

If many genes depart from the trend, the expression is clearly poorly estimated.

In Supplementary Figure 1a, we show that, on the human dataset, the dispersion plot is close to the expected one.

This confirms that the estimation step was correctly carried out.

Second, we plotted the p-value distribution.

The p-value is expected to be uniform under the null hypothesis, and close to 0 under the alternative hypothesis.

The supplementary figure 1b confirms that we observe a mixture of the two distributions on the human dataset.

===

2. Line 121-122:

If the method’s performance does not depend on the values of HMM parameters, is HMM the optimal computational framework to use?

This lack of dependency may suggest a problem with the assumption of the process based on nucleotide-to-nucleotide transition, a problem with data sensitivity (low depth) etc.

 ==> Very true, although one could argue that HMM is robust in this context, which is probably much desired.

 Of note, the HMM method does a good job at finding moderately, differentially expressed small RNAs: our results show it.

 The main advantage with HMM is that it comes with the theoretical background (we exactly know what we optimize), and algorithms.

 That said, HMM is maybe not adapted to very long, little expressed differentially expressed small RNAs.

 This is why we complement it with two other methods.

 IR, for instance, is better when finding long RNAs: among the 10 longest candidates for the human dataset, 9 are provided by the IR method.

 We added this in the text:

===

Ideally, each method is made to complement each other.

The reason is that the expression profile of the small RNAs are very diverse: miRNAs are expected to pile up in short stacks, whereas piRNAs spread over thousand of nucleotides.

We specifically chose the HMM method, because it can detect short, moderately differentially expressed RNAs.

In turn, the IR method can detect longer patterns.

===

3. In the evaluation section, precision and recall are shown only as barplots. Would it make sense to present curves based on the ranked lists of predictions?

 ==> True. We changed barplots to ROC curves in Figure 5. We also changed the description accordingly.

===

The results of each tool as a receiver operating characteristic (ROC) curve, given in Figure 5.

The ROC curve provides the sensitivity against (1 $-$ specificity).

The area under the curve (AUC) gives the probability that a tool ranks a randomly chosen true candidate higher than a randomly chosen wrong one.

The figure shows that srnadiff clearly outperforms other tools.

===

4. Using previously annotated sRNAs as gold standard ("truth set") may be appropriate as a first-level evaluation, but it is not clear how much srnadiff adds to the basic analysis of pre-annotated sRNAs.

It would be important to address the predictions of differentially expressed sRNAs outside of basic annotations, overlaps of these predictions between the three tested methods; and to discuss possible biological relevance and examples of these predictions: for example, potential new miRNAs or piRNAs (hopefully confirmed by the presence of typical sequence patterns), tRNA fragments etc.

 ==> We added a new section in order to answer this question, and the following one.

 Briefly, we added new databases (described in the Methods section), and tried to discover new miRNAs (using miRDeep2) and piRNAs (considering region size and distribution of the reads).

 We reduced the number of unannotated differentially expressed regions to 101.

5. Lines 292-294: For a potential user, it would be important to know more about the new regions predicted by srnadiff but not by the ShortStack method. How many of these regions are previously annotated and how many are novel? How these novel regions are distributed between the known functional genomic elements according to a standard annotation of the genome: introns, exons, enhancers etc? What is the length distribution of these novel regions?

It would also be important to show examples of newly predicted regions of potential biological interest, similar to the previous comment.

 ==> The new section mentioned in the previous paragraph also addresses this point.

 Briefly, we added a plot showing genomic locations and another on showing the region sizes of the differentially expressed regions.

 We organized the plots, so that the contribution of each tool is visible.

 Finally, we added in Supporting Information two screenshots displaying differentially expressed regions, found by srnadiff and no other tool.

Minor comments:

1. Although the goal is to precisely define sRNA positioning in the genome, the 1 bp resolution of read counts may result in lower read counts and lower sensitivity of differential calls by DESeq.

 ==> This is true for messenger RNA sequencing but not exactly for sRNA. We added the following paragraph in the text.

===

This method is expected to give the same results as the annotate-then-identify method for all the sRNAs with a very sharp profile, such as miRNAs and tRFs.

Contrary to messenger RNAs, the reads spans the whole transcript.

Supposing that a sRNA, such as a mature miRNA, is sequenced x times, the annotate-then-identify will quantify an expression of x for this feature.

Our method will also find the same expression of x, because each nucleotide has a sequencing depth of x.

The only difference is that we will likely have identical counts for each nucleotide of the miRNA.

This is why, in our implementation, we collapse lines of the count matrix that have exactly the same counts into a unique row, because the counts actually describe the same object.

===

In a similar vein, what is a reasonable threshold of p-value (line 126) at which a genomic position is omitted from statistical consideration? Was it estimated in a rigorous way based on real data?

 ==> In the line you mention, there is no threshold of p-value.

 Actually, the HMM needs a p-value for each nucleotide of the genome, and even a non significant p-value of 0.9 can be correctly interpreted by the HMM.

 However, we did use a threshold for the coverage: by default, there should be at least one sample with at least 10 reads.

 The aim here was to save computation time, with (almost) no decrease of sensibility.

 We used this (admittedly arbitrary) threshold based on our experience.

 A region which is covered by less than 10 reads, even in the most expressed sample, is not likely to be reliably called differentially expressed.

 This parameters can be easily changed by the user, though.

2. English needs minor corrections and stylistic editing.

 ==> Thanks! We attentively read the manuscript, and tried to correct all the mistakes.

Reviewer #5: The work proposed a method called “srnadiff” in order to find differentially expressed sRNA, without annotation (the annotation is optional).

The authors divide the proposed approach into two steps. The first step comprises applying some methods like HMM (hidden Markov model), IR (Irreducible regions), annotation (as optional), etc. in order to produce genomic intervals that are potential differentially expressed regions.

The second step comprises clustering the samples and applying the DESeq2 method for the identification of differential expression.

Essentially, the problem addressed by the paper is not one of detecting differential expression, in numerical or statistical terms, but rather one of mapping, i.e., the contribution of the work is essentially in step 1.

Therefore, the manuscript presents an R package for the detection of differentially expressed smallRNAs. The text is difficult to understand and does not present clearly with tables and figures (see specific comments below). The authors do not sustain the reasons their results are better (identifying more differentially expressed sequences does not exactly mean that the tool is better), perhaps this difficulty in making clear is because of the way the comparison was done.

The work is interesting and applied in a significant bioinformatics research context. Some points could be better presented which could improve the quality of the work.

Major:

Regarding the presentation of the method, it would be very important to improve the presentation of step 1 by clearly contextualizing how the different adopted approaches are integrated to perform the clustering of genomic regions. Making clear the adopted criteria, information flow and parameters considered, among other information.

 ==> First, we reorganized the manuscript to improve the flow.

 The description of the segmentation methods are now described in the "Step 1" section.

 We also added some contextualization: for instance, before the description of Step 1 and 2, we added:

===

Briefly, the first step includes several, independent methods that detect putative differentially expressed regions.

The second step considers all the regions found by the previous step, and chooses the best ones.

===

 We also added inputs and outputs, and parameters used, for each method.

 Figure 1 has also been modified to include parameters, and a little cartoon of the data which are produced by each tool (coverages and intervals).

Explain and include references to which methods and implementations were adopted for the HMM and IR.

 ==> We did not make it clear that we implemented the method from scratch.

 We added "The algorithm has been implemented from scratch in C++." in the corresponding paragraphs, in the "Strategy" section, and we cited again the reference of the IR method.

 Concerning the Viterbi algorithm, being a textbook algorithm, there is no clear reference to cite.

The acronym snoRNAs is used but not previously presented as piRNAs for example.

 ==> Thanks, we corrected that in the introduction.

The method describes expression identification with only RNA-Seq data without reference, but in Figure 1 the first step uses a BAM (output file mapping reads to a reference).

 ==> This is a major misunderstanding, and we are sorry that we did not make it clear.

 The method does not need a reference annotation (but can use it, if available), but does need a reference genome.

 We tried to correct this, especially in the Abstract, where we mentioned "reference sequences of sRNAs".

 The new text is now:

===

To the extent of our knowledge, srnadiff is the first tool that detects differentially expressed sRNAs without the use of external information, such as genomic annotation or additional sequences of sRNAs.

===

 Moreover, we added to following lines in the introduction.

===

In this work, we will suppose that a reference genome is available.

Although many organisms are still not assembled, most model and many non-model organisms are available, and the sequencing of the remaining organisms progress at a fast pace.

===

In step 2, the proposed work applies the DESeq2 method as a method for identifying differential expression. The proposed approach is available from a Bioconductor package. The suggestion to the authors is to make available to use it as a library and apply other methods for differential expression detection. An interesting proposal would be to apply the methods together, as proposed in this work:

https://journals.plos.org/plosone/article?id=10.1371/journal.pone.0190152

 ==> We usually think that, for a package, adding several tools is a bad practice.

 First, it increases the number of dependencies, and makes a package more difficult to maintain.

 Second, it confuses the user, because she/he does not know which tool is best.

 We usually prefer benchmarking the tools, and finally choose the best one.

 The example that you refer is a pipe-line, with different purpose in mind.

 Yet, the reviewer is always right, so we added the tools.

 We considered the four tools which were mentioned in the article, but discarded NOISeq, because it does not seems maintained any longer.

 We added a new paragraph, called "Computation of the p-values" in the text.

 The text is:

===

p-values at computed twice in the process.

First, they are computed for each position of the genome in the HMM strategy.

Second, candidate regions are tested for differential expression, and given a p-value.

By default, DESeq2 is used to perform the two steps.

However, for the sake of completeness, we made two other widely used packages available for the user: edgeR, and baySeq.

The alternative package can be chosen by specifying the parameter diffMethod in the srnadiff function.

===

 We also added Figure 8, which compares the 3 packages on the simulated dataset, and shows that the DESeq2 (default) method gives best results.

The tool itself is well done and seems to me to be really necessary, even for computational efficiency, but the way it is presented can be much improved, despite the corrections already stated in revision 1.

Regarding the results, in the presentation in Fig. some works are mentioned (article 1, article 2, usual differential expression calling method). It is important to contextualize these works in the text, justify the choice of these papers and their characteristics so that it is possible to compare them contextually with the proposed approach.

 ==> We are sure about the Figure you mentioned.

 Anyway, we now explicitly present articles 1 and 2 as reviews on plants and animals on small RNAs.

===

These sRNAs include the well-known microRNAs, but also tRNA-derived RNA fragments (tRFs), small interfering RNAs (siRNAs), Piwi-associated RNAs (piRNAs), small nucleolar RNAs (snoRNAs), which have been extensively studied in plants and animals (see [1, 2] for reviews on sRNAs on both kingdoms).

===

 In the "Benchmark" section, we specified that we chose three random studies, so that they represent the diversity of the small RNA approaches.

===

For the published datasets, we randomly choose several ones from the vast literature, with few criteria.

First, there should be replicates, second the differentially expressed elements should be accessible as supplementary data of the article.

We then wanted to encompass a variety of model organisms: Homo sapiens, Arabidopsis thaliana and Drosophila melanogaster.

The analysis used by the three articles are also quite different, as well as the sequencing machines.

===

 Moreover, for each dataset, we present the method which has been used by in the original work.

 For the human dataset: 

====

In the first paper, the authors mapped the reads with bowtie on miRBase.

The miRNAs were tested for differential expressed with edgeR.

In the second paper, the authors mapped the reads with Novoalign on the reference genome.

They downloaded piRNA and snoRNA annotations, and quantified their expression with the BedTools.

They were tested for differential expressed with edgeR.

====

 For the thale cress dataset:

====

The authors mapped the reads with RazerS on the reference genome.

Reads co-localizing with known miRNAs were kept for further analysis.

The authors then looked for other, unknown, putative miRNAs.

Other reads were compared to known cDNAs, miRNAs, tRNAs, rRNAs, snoRNAs, and other RNA sequences, and matching reads were excluded.

Remaining reads were then given to miRDeep and miRCat, two miRNA finding tools.

The new set was added to the previous, known set of miRNAs.

Differential expression analysis was performed with a X2 test.

====

 For the fruit fly dataset:

====

The authors mapped the reads with bowtie on the reference genome.

Detection of new miRNAs, and quantification of known and new miRNAs, was performed with miRDeep2, using data from miRBase.

Micro RNAs were tested for differential expressed with edgeR.

====

Fig. 4 can be better presented and contextualized. It is important that the authors make clear the numbers and definitions of what they considered as True Positive and True Negative for each of the datasets. In addition, it is important that they present the Precision (TP/(TP+FP)) and Recall (TP/(TP+FN)) rates in order to make clear the improvement provided by the proposed method. Clarify that the method recovers regions with differential expression and with precision, i.e., without an excessive number of false positives. Also, correct the y-axis labeling of the figures. It would interest to make clear what these results show. It is important to explain and contextualize where this better result can be visualized and why it is described as “better”.

 ==> Thank you for the advice, we tried to follow it.

 We clarified the context, and we focused on recall, but no precision, because we have no exhaustive true set (and it is very complicated to obtain it).

 However, we mentioned that a precision will be assessed in the simulated dataset.

 Of note, we did not mention that a tool was "better", but that some tools (such as srnadiff) missed less putative regions.

 We explained this in the "Benchmarking" section.

===

Of note, it is not possible to assess the number of false positives, \\textit{i.e.} the regions detected by the evaluated tools, on real-life datasets, because there exists no method that finds all the differentially expressed small RNAs.

Therefore, metrics such as specificity or precision cannot be collected on these dataset.

Our aim, here, is to test whether the evaluated tools are able to detect all the regions that have been classified as differentially expressed by the reference methods.

The simulated dataset, even though it is imperfect because it poorly reproduces the erratic distribution of the reads observed in real-life datasets, can assess both the number of false positives, and false negatives.

This is why we added a fourth, synthetic dataset, to the real-life ones.

===

 Before Figure 4, we clarified what we called "true positive".

===

Figure 4 shows the recall of the different methods.

The recall (or sensitivity) is defined as TP/(TP+FN), where TP is the number of true positives, and FN is the number of false negatives.

In our benchmark, a true positive is region which has been detected as differentially expressed by a reference set (here, the miRNAs published in the articles), such that at least 80\\% of the nucleotides overlap with a region detected as differentially expressed by srnadiff (or other evaluated tools such as derfinder, and ShortStack).

Conversely, the false negatives are the regions which has been detected as differentially expressed by a reference set, which do not overlap (with at least 80% of the nucleotides) with a region detected as differentially expressed by the evaluated tools.

===

Fig. 3 refers to the comparison tools as source, 4 tool, 5 and 6 method.

 ==> Yes! We used "tool" to be more consistent.

Fig. 6, Y-axis should be better identified what represents “value”?

 ==> Yes! We mentioned that the top plot was "time (in s)", and the bottom "space (in Mb)".

The p-value was defined for all methods, however the fold-change is mentioned only for synthetic dataset. It is important to make clear the adopted p-value and fold-change for all methods and datasets.

 ==> We already mentionned the p-value thresholds in the "Benchmark" section:

===

srnadiff was run with no annotation, and an adjusted p-value threshold of 5%.

We also run derfinder on the same datasets, with a q-value of 10%.

We used a third method, which first clusters the reads with ShortStack (comparing several clustering methods is out of the scope of this article), quantified the expression of the regions found by ShortStack with featureCounts, tested for differential expression with DESeq2, and kept the regions with an adjusted p-value of at most 5%.

[...]

The reason why we chose a q-value of 10% for derfinder, instead of 5%, is that the statistics produced by derfinder is significantly more conservative, and it produces much less predicted regions than other approaches.

For a fair comparison, we decided to lower the stringency for this tool.

===

 We added that no fold change was applied.

 Of note, srnadiff now provides the log2FC of the called regions.

The preprocessed dataset used to generate the results was not made available by the authors. It is recommended that the dataset used be made available that allows the replication of the results by the research community.

 ==> Very true.

 We uploaded all the BAM files, the experimental design files, to a public dataverse with a persistent DOI: https://doi.org/10.15454/0DCIGO

 This information has been added in Supporting Information.

Minor:

- Fig. 2 low resolution.

 ==> Fixed!

---

## [Decision Letter · Decision Letter 2]

7 Jul 2021

PONE-D-20-29322R2

Finding differentially expressed sRNA-Seq regions with srnadiff

PLOS ONE

Dear Dr. Matthias,

Thank you for submitting your manuscript to PLOS ONE. After careful consideration, we feel that it has merit but does not fully meet PLOS ONE’s publication criteria as it currently stands. Therefore, we invite you to submit a revised version of the manuscript that addresses the points raised during the review process.

We look forward to receiving your revised manuscript.

Kind regards,

J Francis Borgio, Ph.D.,

Academic Editor

PLOS ONE

Journal Requirements:

Reviewers' comments:

Reviewer's Responses to Questions

**Comments to the Author**

1. If the authors have adequately addressed your comments raised in a previous round of review and you feel that this manuscript is now acceptable for publication, you may indicate that here to bypass the “Comments to the Author” section, enter your conflict of interest statement in the “Confidential to Editor” section, and submit your "Accept" recommendation.

Reviewer #2: (No Response)

Reviewer #4: (No Response)

Reviewer #5: All comments have been addressed

2. Is the manuscript technically sound, and do the data support the conclusions?

Reviewer #2: (No Response)

Reviewer #4: Yes

Reviewer #5: Yes

3. Has the statistical analysis been performed appropriately and rigorously? 

Reviewer #2: (No Response)

Reviewer #4: Yes

Reviewer #5: Yes

4. Have the authors made all data underlying the findings in their manuscript fully available?

Reviewer #2: (No Response)

Reviewer #4: Yes

Reviewer #5: Yes

5. Is the manuscript presented in an intelligible fashion and written in standard English?

Reviewer #2: (No Response)

Reviewer #4: No

Reviewer #5: Yes

6. Review Comments to the Author

Reviewer #2: (No Response)

Reviewer #4: The authors addressed many of reviewers’ concerns: discussed the unusual application of DESeq, added more detailed results of performance evaluation, described differences of these results from simpler standard methods based on reference annotations, showed examples of possible novel small RNAs identified as differentially expressed etc. Although the number of detected new intergenic small RNAs outside of standard annotation is relatively small (~100), this pipeline may still be worth reporting.

My main current concern is the language, especially in the new edits. These are just a few examples:

P. 4 – need clarifying:

‘Ideally, each method is made to complement each other. The reason is that the expression profile of the small RNAs are very diverse’

P. 6: poor language:

‘The algorithm has been implemented from scratch in C++.’

P.8: a few passages need editing:

‘Concerning the published datasets, we randomly choose them from the vast literature, with few criteria… The analysis used by the three articles are also quite different, as well as the sequencing machines.’

‘The simulated dataset, even though it is imperfect because it poorly reproduces the erratic distribution of the reads observed in real-life datasets, can assess both the number of false positives, and false negatives. This is why we added a fourth, synthetic dataset, to the real-life ones.’

P. 12: Unclear: does this mean that no novel piRNAs were detected by the new method?

‘This seems to indicate that no differentially expressed piRNA was missed.’

The new supplemental figure with genomic tracks of the examples of new detected small RNAs also needs additional work. As of now, it is just a screenshot from IGV, where it is difficult to read both vertical and horizontal scales, and there are many unnecessary image elements that should be removed.

Reviewer #5: The work is interesting and significantly applied context of bioinformatics research. All points raised have been addressed appropriately, leading to significant improvement in the manuscript's quality and the availability of the data used.

7. PLOS authors have the option to publish the peer review history of their article (what does this mean?). If published, this will include your full peer review and any attached files.

Reviewer #2: No

Reviewer #4: No

Reviewer #5: No

---

## [Author Response · Author response to Decision Letter 2]

20 Jul 2021

Dear Reviewer,

Again, we thank you very much for the time you spent to improve our manuscript.

Please find hereafter the answers to your suggestions.

Best regards,

Ignacio González and Matthias Zytnicki.

My main current concern is the language, especially in the new edits.

==> We carefully read the text again, and corrected a few typos and errors.

These are just a few examples:

P. 4 – need clarifying:

‘Ideally, each method is made to complement each other. The reason is that the expression profile of the small RNAs are very diverse’

 ==> We replaced the sentences with the following ones:

===

Indeed, expression profiles are very specific to each of the sRNA classes: miRNAs pile up in short stacks, whereas piRNAs spread over thousand of nucleotides.

In order to cope with this diversity, we found that two different methods yielded better results than a unique one.

The HMM method efficiently detects short, moderately differentially expressed RNAs.

The IR method can detect longer patterns.

===

P. 6: poor language:

‘The algorithm has been implemented from scratch in C++.’

 ==> In the HMM section, we replaced the sentence with:

===

We implemented this Viterbi algorithm, optimized for our purposes, in C++.

===

 ==> In the IR section, we replaced the sentence with:

===

We implemented the algorithm in C++, using the description from [22].

===

P.8: a few passages need editing:

‘Concerning the published datasets, we randomly choose them from the vast literature, with few criteria… The analysis used by the three articles are also quite different, as well as the sequencing machines.’

 ==> We replaced the sentences with:

===

We selected several datasets meeting the following criteria.

First, the experimental design should include replicates in each condition.

Second, the differentially expressed elements found by the authors should be accessible as supplementary data of the article.

Third, we wanted to include a variety of model organisms: Homo Sapiens, Arabidopsis thaliana and Drosophila melanogaster.

Fourth, the sequencing machines should be different, in order to include the diversity of the machines in our benchmark.

Last, the analysis pipe-lines should also be different.

===

‘The simulated dataset, even though it is imperfect because it poorly reproduces the erratic distribution of the reads observed in real-life datasets, can assess both the number of false positives, and false negatives. This is why we added a fourth, synthetic dataset, to the real-life ones.’

 ==> We replaced the sentences, with the following, shorter, one, which fits better right after the previous paragraph:

===

In order to assess the number of false positives, we added a last, simulated, dataset.

===

P. 12: Unclear: does this mean that no novel piRNAs were detected by the new method?

‘This seems to indicate that no differentially expressed piRNA was missed.’

 ==> True, the sentence was unclear.

 In the previous paragraph, we showed that, among the unannotated differentially expressed regions, 30 were in fact piRNAs (using an additional database).

 In the paragraph mentioned by the reviewer, we showed that the remaining unannotated differentially expressed regions were short, with no bias U/A bias on the first/tenth position.

 So, we changed the sentence to:

===

This seems to indicate that the unannotated differentially expressed regions are probably not piRNAs.

===

The new supplemental figure with genomic tracks of the examples of new detected small RNAs also needs additional work. As of now, it is just a screenshot from IGV, where it is difficult to read both vertical and horizontal scales, and there are many unnecessary image elements that should be removed.

 ==> The image has been modified, so that every information is easily readable.

 Unnecessary information has been removed.

---

## [Decision Letter · Decision Letter 3]

3 Aug 2021

Finding differentially expressed sRNA-Seq regions with srnadiff

PONE-D-20-29322R3

Dear Dr. Matthias,

We’re pleased to inform you that your manuscript has been judged scientifically suitable for publication and will be formally accepted for publication once it meets all outstanding technical requirements.

Kind regards,

J Francis Borgio, Ph.D.,

Academic Editor

PLOS ONE

Additional Editor Comments (optional):

Reviewers' comments:

Reviewer's Responses to Questions

**Comments to the Author**

1. If the authors have adequately addressed your comments raised in a previous round of review and you feel that this manuscript is now acceptable for publication, you may indicate that here to bypass the “Comments to the Author” section, enter your conflict of interest statement in the “Confidential to Editor” section, and submit your "Accept" recommendation.

Reviewer #4: All comments have been addressed

2. Is the manuscript technically sound, and do the data support the conclusions?

Reviewer #4: Yes

3. Has the statistical analysis been performed appropriately and rigorously? 

Reviewer #4: Yes

4. Have the authors made all data underlying the findings in their manuscript fully available?

Reviewer #4: Yes

5. Is the manuscript presented in an intelligible fashion and written in standard English?

Reviewer #4: Yes

6. Review Comments to the Author

Reviewer #4: My final comments were fully addressed and the manuscript's clarity and style improved.

7. PLOS authors have the option to publish the peer review history of their article (what does this mean?). If published, this will include your full peer review and any attached files.

Reviewer #4: No

---

## [Editor Report · Acceptance letter]

11 Aug 2021

PONE-D-20-29322R3 

Finding differentially expressed sRNA-Seq regions with srnadiff  

Dear Dr. Zytnicki:

I'm pleased to inform you that your manuscript has been deemed suitable for publication in PLOS ONE. Congratulations! Your manuscript is now with our production department. 

Kind regards, 

on behalf of

Dr. J Francis Borgio 

Academic Editor

PLOS ONE